# Reversal of ApoE4-induced recycling block as a novel prevention approach for Alzheimer's disease

Xunde Xian[1,2†]*, Theresa Pohlkamp[1,2†], Murat S Durakoglugil[1,2], Connie H Wong[1,2], Jürgen K Beck[3], Courtney Lane-Donovan[1,2], Florian Plattner[2,4], Joachim Herz[1,2,5,6]*

[1]Department of Molecular Genetics, University of Texas Southwestern Medical Center, Dallas, United States; [2]Center for Translational Neurodegeneration Research, University of Texas Southwestern Medical Center, Dallas, United States; [3]Jkb Consult Inc SPRL, Brussels, Belgium; [4]Department of Psychiatry, University of Texas Southwestern Medical Center, Dallas, United States; [5]Department of Neuroscience, University of Texas Southwestern Medical Center, Dallas, United States; [6]Department of Neurology and Neurotherapeutics, University of Texas Southwestern Medical Center, Dallas, United States

*For correspondence:
Xunde.xian@utsouthwestern.edu
(XX);
joachim.herz@utsouthwestern.edu
(JH)

†These authors contributed
equally to this work

Competing interests: The
authors declare that no
competing interests exist.

Reviewing editor: Hugo J
Bellen, Baylor College of
Medicine, United States

**Abstract** ApoE4 genotype is the most prevalent and also clinically most important risk factor for late-onset Alzheimer's disease (AD). Available evidence suggests that the root cause for this increased risk is a trafficking defect at the level of the early endosome. ApoE4 differs from the most common ApoE3 isoform by a single amino acid that increases its isoelectric point and promotes unfolding of ApoE4 upon endosomal vesicle acidification. We found that pharmacological and genetic inhibition of NHE6, the primary proton leak channel in the early endosome, in rodents completely reverses the ApoE4-induced recycling block of the ApoE receptor Apoer2/Lrp8 and the AMPA- and NMDA-type glutamate receptors that are regulated by, and co-endocytosed in a complex with, Apoer2. Moreover, NHE6 inhibition restores the Reelin-mediated modulation of excitatory synapses that is impaired by ApoE4. Our findings suggest a novel potential approach for the prevention of late-onset AD.
DOI: https://doi.org/10.7554/eLife.40048.001

## Introduction

Over the course of the last 30 years, we have learned much about the genetics of Alzheimer's disease (AD), yet the pathological mechanisms that cause the onset of the disease and that define its progression culminating in massive neurodegeneration and debilitating dementia remain poorly understood. We know that amyloid-β (Aβ), an aggregation-prone proteolytic processing product that consists of juxtamembrane sequences and approximately 2/3 of the transmembrane segment of the larger amyloid precursor protein (APP), plays a central role early on, while during later stages of the disease – through mechanisms that are completely obscure – the microtubule-associated protein τ begins to form intraneuronal aggregates, that is the neurofibrillary tangles, that can spread transsynaptically (*Selkoe and Hardy, 2016*). It is this τ-aggregation, not the amyloid plaques, which occur early in the disease process, that is thought to be primarily responsible for the neuronal cell death and brain mass loss and that most closely tracks with dementia progression. Before they become visible as the telltale plaques and tangles that cement the AD diagnosis for the pathologist, Aβ and τ are present as smaller oligomeric aggregates that can directly and profoundly impair neuronal functions, by disrupting synaptic $Ca^{2+}$ homeostasis (*Kuchibhotla et al., 2008*). The resulting synaptic

**eLife digest** Alzheimer's disease is a degenerative condition that destroys connections between brain cells leading to memory loss, confusion and difficulties in thinking. Apolipoprotein E is a protein that carries fatty substances called lipids and cholesterol around the brain, and plays an important role in repair mechanisms. There are three major forms of Apolipoprotein E, and individuals who carry a version known as ApoE4 are up to 10 times more likely to develop Alzheimer's disease than those who carry other variations.

In nerve cells, or neurons, Apolipoprotein E binds to a specific family of receptors. One of these receptors, called Apoer2, is found in the synaptic gap between neurons, where it regulates their activities. Both Apolipoprotein E and Apoer2 are taken into the cell within compartments known as endosomal vesicles. Usually, the Apoer2 receptor is quickly recycled back to the surface of the cell, but this recycling process is delayed in individuals with the ApoE4 version of Apolipoprotein E.

Apoer2 is just one of many different receptors on the surface of neurons that are taken into vesicles before being recycled back to the cell surface. The fluid inside these vesicles becomes progressively more acidic as they move through the cell. This process helps to control the interaction of these receptors with their binding partners and to regulate their movement and recycling. Here, Xian, Pohlkamp et al. investigated whether changing the acidity of vesicles in rat neurons could overcome the block in recycling Apoer2 – and other receptors that travel with Apoer2 in the same compartments – in the presence of ApoE4.

A protein called NHE6 is embedded in the membrane of vesicles called early endosomes and acts to make the vesicles less acidic. Xian, Pohlkamp et al. used drugs to block the activity of NHE6, which led to the vesicles becoming more acidic and allowed Apoer2 to be recycled faster. Using a genetic approach known as siRNA knockdown to decrease the amount of NHE6 produced in neurons also had a similar effect on Apoer2 recycling.

Together these findings suggest that drugs that make vesicles in neurons more acidic may have the potential to help prevent individuals that carry the ApoE4 protein from developing Alzheimer's disease. Current drugs that target NHE6 also affect other molecules, which can often lead to side effects. A next step will be to develop tailor-made, small molecule drugs that can enter the brain efficiently and selectively block NHE6.

DOI: https://doi.org/10.7554/eLife.40048.002

dysfunction is widely thought to represent the earliest stage of the AD pathogenic process and to be a major cause of its clinical manifestation as mild cognitive impairment (MCI) (*Palop and Mucke, 2010*; *Shankar and Walsh, 2009*).

Although AD almost certainly involves early Aβ accumulation, only a vanishingly small number of individuals suffering from the vicious and dominant early-onset form of AD carry mutations in APP (*Campion et al., 1999*). These missense mutations invariably increase Aβ production and lead to early plaque deposition, while a similarly small number of people have a genetic variation in APP that lowers Aβ production and hence protects from the much more frequent late-onset form of AD (LOAD) (*Jonsson et al., 2012*). Numerous other genetic determinants contribute to the bulk of LOAD, which typically develops after the 6th decade. The most important of these is Apolipoprotein E ε4 (ApoE4) genotype (*Corder et al., 1993*; *Strittmatter et al., 1993*).

ApoE is a lipid and cholesterol carrying protein that is primarily produced by the liver and is responsible for plasma lipid homeostasis (*Mahley, 1988*). It occurs in three major isoforms in humans known as ApoE2, ApoE3 and ApoE4, with ApoE3 being the most frequent allele (~77% homozygosity) followed by ApoE4 (~15 – 20% allele frequency) which is present in >50% of LOAD (*Liu et al., 2013*). The effect of ApoE4 on Aβ accumulation through impaired Aβ turnover, increased aggregation and thus plaque formation is allele dosage-dependent and this can partly explain its effect on the earlier age of disease onset (*Corder et al., 1993*). However, ApoE4 can independently impair synapse function and $Ca^{2+}$ homeostasis by disrupting the endocytic transport and recycling of synaptic ApoE receptors and the excitatory AMPA and NMDA type glutamate receptors that are regulated by those ApoE receptors and that are consequently trapped with them in the same vesicles (*Chen et al., 2010*).

Most ApoE receptors, which are all members of the low-density lipoprotein (LDL) receptor gene family, are expressed in the brain and several are intrinsic components of excitatory synapses where they are present in the presynaptic and postsynaptic compartments (reviewed in *Lane-Donovan et al., 2014*; *Pohlkamp et al., 2017*). Of these, ApoE receptor-2 (Apoer2, a.k.a. LRP8) is the best characterized. It is present both pre- as well as postsynaptically where it primarily functions as a receptor for Reelin (*Bal et al., 2013*; *Beffert et al., 2005*; *Lane-Donovan and Herz, 2017*). Reelin is a large secreted protein that is essential for the formation of cortical layers during embryonic brain development where it serves as a guidance molecule in the regulation of neuronal migration (*D'Arcangelo et al., 1995*; *Del Río et al., 1997*). As the brain continues to develop and mature postnatally, its expression pattern changes and Reelin is now produced by a subset of GABAergic interneurons that are interspersed throughout the neocortex and the hippocampus (*Alcántara et al., 1998*; *Pesold et al., 1998*; *Pohlkamp et al., 2014*). In the adult brain, this secreted Reelin now functions as a neuromodulator by signaling through Apoer2 and its closely related family member Vldlr to activate Src-family tyrosine kinases directly in the synapse, which results in increased $Ca^{2+}$ influx through NMDA receptors and thus the robust elevation and maintenance of synaptic potentiation (*Chen et al., 2005*; *Hiesberger et al., 1999*; *Wasser and Herz, 2017*). This is the key event in the maintenance of synaptic homeostasis that is impaired by ApoE4 and which occurs independent of Aβ accumulation (*Chen et al., 2005*). Indeed, ApoE4-specific alterations in brain structure have been found in <2 year old children (*Dean et al., 2014*; *Shaw et al., 2007*).

The molecular basis by which ApoE4 causes the disruption of normal endosomal vesicle transport and recycling is most likely the result of its propensity to unfold and assume a 'molten-globule' conformation upon entering an acidic environment (*Morrow et al., 2002*). ApoE4 differs from ApoE3 by a single amino acid, which alters its isoelectric point to coincide with the pH of ~6.5 that is present in the early endosome (*Casey et al., 2010*; *Ordovas et al., 1987*). We hypothesized that this isoelectric charge neutralization would make ApoE4 prone to aggregation, which could be the molecular basis for the ApoE4-induced and gene dosage-dependent recycling defect.

pH in the early endosome is maintained by the opposing functions of the proton pump, which decreases vesicular pH, and the $Na^+/H^+$ exchanger NHE6, which increases it (*Fuster and Alexander, 2014*). Here, we have investigated the role of NHE6 inhibition as a means of lowering endosomal pH, away from the isoelectric point of ApoE4. We found that this simple pharmacological intervention releases the endosomal ApoE4 block, restores the normal trafficking of ApoE receptors and glutamate receptors in neurons and corrects the functional defects in vitro and in vivo. These findings suggest NHE6 inhibition as a novel rational therapeutic approach for reversing the AD risk imposed by ApoE4.

## Results

### ApoE interacts and colocalizes with Apoer2 in neurons

ApoE generically interacts with cysteine-rich ligand binding-type repeats that are ubiquitously present in all LDL receptor family members (*Blacklow, 2007*). We first investigated the interaction between ApoE and its ligand-receptor Apoer2 using a solid phase interaction assay. For this purpose, we used naturally secreted ApoE particles containing the three common ApoE isoforms in humans (ApoE2, ApoE3 and ApoE4). We found that ApoE3 and ApoE4 strongly interact with Apoer2, whereas ApoE2 binding was much weaker (*Figure 1A and B*). These results are consistent with the established similarly high-affinity binding of ApoE3 and ApoE4 to the LDL receptor and the 100-fold reduced affinity of ApoE2 (*Rall and Mahley, 1992*; *Weisgraber, 1994*). Thus, Apoer2 is a high-affinity ApoE receptor and immunofluorescence analysis of primary neurons treated with GFP-tagged ApoE3 accordingly showed co-localization of Apoer2 and ApoE3 in endosomes (*Figure 1C*, *Videos 1* and *2*).

### ApoE4 selectively reduces cell surface expression of neuronal Apoer2

We have previously reported that the presence of receptor binding competent ApoE4 particles at physiological concentrations impairs the recycling and consequently surface expression of neuronal Apoer2 (*Chen et al., 2010*). To test whether other unrelated receptors that do not interact with ApoE may be also affected in their plasma membrane expression and recycling properties, we

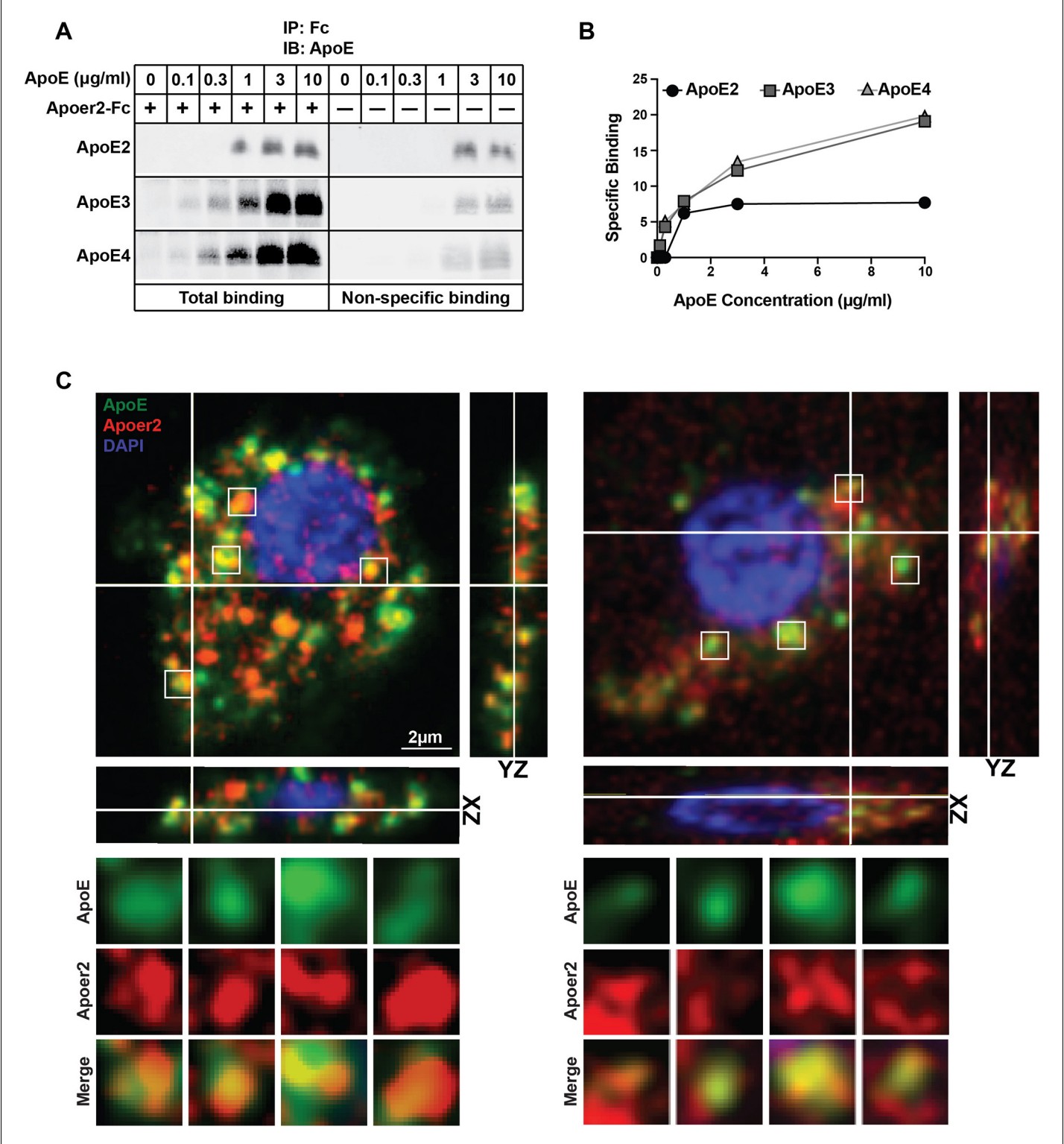

**Figure 1.** Binding of ApoE Isoforms to Apoer2. (**A and B**) ApoE isoforms interact with ApoE receptor 2 (Apoer2) as tested by co-immunoprecipitation. ApoE3 and ApoE4 bind Apoer2 with similar affinity, whereas ApoE2 binding to Apoer2 is poor. ApoE-conditioned media (0, 0.1, 0.3, 1, 3 and 10 µg/ml ApoE) were incubated with Apoer2-Fc (secreted Apoer2 ectodomain fused to Fc) bound to protein-G beads and pulled down to perform immunoblotting for ApoE. Representative immunoblot images (**A**) and quantification (**B**) are shown. (**C**) Apoer2 co-localizes with ApoE in primary neurons. Primary cortical neurons were infected with lentiviral mCherry-Apoer2 (red) and subsequently treated with ApoE3-GFP-conditioned media (green). A single plane of a z-stack is shown with the orthogonal xz- and yz-views as indicated. White lines indicate the vertical and horizontal cuts.
*Figure 1 continued on next page*

*Figure 1 continued*

Boxed vesicles are shown enlarged in the panels below labeled ApoE, Apoer2 and Merge. Additionally 3D movies of the cells are provided online (*Videos 1* and *2*).

DOI: https://doi.org/10.7554/eLife.40048.003

The following source data is available for figure 1:

**Source data 1.** Binding of ApoE Isoforms to Apoer2.

DOI: https://doi.org/10.7554/eLife.40048.004

performed the cell surface biotinylation experiments outlined in *Figure 2A*. Briefly, primary rat cortical neurons were incubated in the absence (*Figure 2B*, lane 1) or presence of cell-derived, naturally secreted recombinant ApoE3 (lane 2) or ApoE4 (lane 3) particles for 1 hr at 37°C, purified Reelin was then added to induce the rapid endocytosis of Apoer2, and after 30 min the cells were transferred to 4°C and washed with ice-cold PBS. Cell surface biotinylation was performed, biotinylated proteins were isolated and detected by immunoblotting. While Apoer2 quickly recycled in the presence of ApoE3, its reappearance on the cell surface was greatly delayed in the presence of ApoE4 (*Figure 2B,C*). By contrast, other endocytic cell surface receptors that do not bind ApoE, such as the insulin receptor (IR) or the transferrin receptor (TfR), or that do not interact with Reelin and therefore do not undergo ligand-induced endocytosis (low-density lipoprotein receptor-related protein 1 (Lrp1) and low-density lipoprotein receptor (Ldlr)), were not significantly affected by the presence of ApoE (*Figure 2B,C*).

## Prolonged retention and activation of Apoer2 by ApoE4

The delayed recycling of Apoer2 was also apparent by the prolonged retention of cell-derived ApoE4 compared to cell-derived ApoE3 in Apoer2-containing intracellular compartments as shown by co-immunoprecipitation of neuronal lysates (*Figure 2D,E*). That this prolonged retention may be caused by the partial unfolding of ApoE4 is further supported by an experiment in which increasing amounts of naturally secreted, receptor binding-competent ApoE4 particles were added to primary cortical neurons and tyrosine phosphorylation of Dab1 was measured. Dab1 binds to the NPxY motif in the cytoplasmic domain of Apoer2 and when the receptors are clustered, for example by interacting with Reelin, Dab1 undergoes trans-phosphorylation on tyrosine residues (*Hiesberger et al., 1999*; *Howell et al., 1997*). We hypothesized that ApoE4 in its molten-globule state, that is in acidic endosomes, might similarly induce receptor clustering in a dose-dependent manner, whereas ApoE3 would not. When we treated primary neurons with ApoE4, Dab1 phosphorylation was indeed increased as expected (*Figure 2F,G*).

## Pharmacological NHE inhibition and acidification of vesicular pH restores Apoer2 trafficking in the presence of ApoE4

The pioneering work of Weisgraber and colleagues revealed the propensity of ApoE4 to become structurally labile and undergo transformation to a molten-globule state in a low pH environment, while ApoE2 and ApoE3 were far more resistant to low pH-induced unfolding (*Morrow et al., 2002*). In addition, we noticed that the isoelectric point (IEP) of ApoE4 lies close to the pH in the early endosome

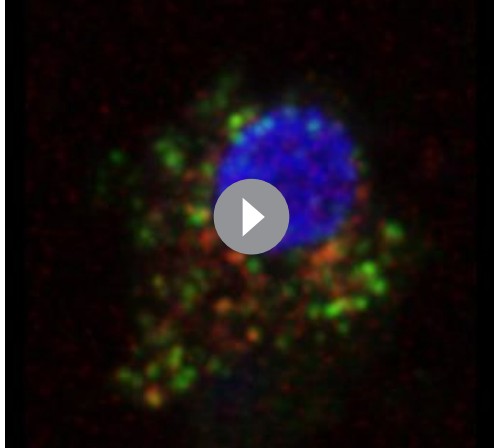

**Video 1.** Supporting material for *Figure 1C*. 3D View of Apoer2 co-localizes with Apoe in primary neurons. N-terminal mCherry-labeled Apoer2 (red) and C-terminal GFP-labeled ApoE3 (green) co-localize intracellularly in primary neurons. Rat primary cortical neurons were infected with lentiviral mCherry-Apoer2 and subsequently exposed to ApoE3-GFP-conditioned media. Confocal microscopy was performed as described in the Materials and methods section.
DOI: https://doi.org/10.7554/eLife.40048.005

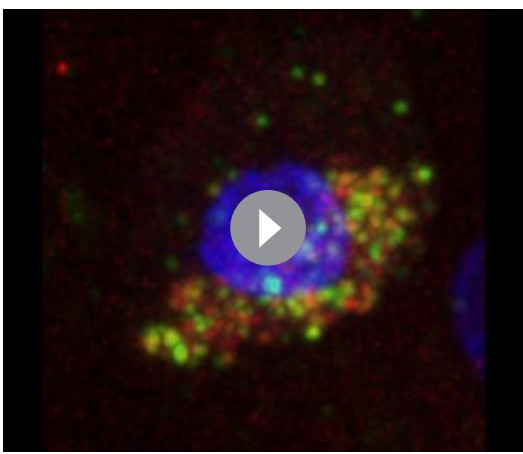

**Video 2.** Supporting material for *Figure 1C*. 3D View of Apoer2 co-localizes with Apoe in primary neurons. N-terminal mCherry-labeled Apoer2 (red) and C-terminal GFP-labeled ApoE3 (green) co-localize intracellularly in primary neurons. Rat primary cortical neurons were infected with lentiviral mCherry-Apoer2 and subsequently exposed to ApoE3-GFP-conditioned media. Confocal microscopy was performed as described in the Materials and methods section.
DOI: https://doi.org/10.7554/eLife.40048.006

(*Figure 3A*). Many proteins are known to lose hydrophilicity near their IEP. Indeed, the first purification of insulin depended upon this biophysical phenomenon (*Wintersteiner and Abramson, 1933*). We thus hypothesized that the structural lability of ApoE4, combined with a reduced solubility in the acidic endosomal environment, might be a driver for the resulting recycling block. pH regulation in the endosome is achieved by a combination of two primary mechanisms: the activity of the proton pump, that is v-ATPase and the exchange of protons for $Na^+$ or $K^+$ through the activity of $Na^+/H^+$ exchangers (NHEs) (*Figure 3B*). Functional disruption of the endosomal NHE6 isoform thus lowers endosomal pH (*Brett et al., 2002*) and would be predicted to restore ApoE4 solubility and thus vesicle trafficking (*Figure 3C*).

NHE6-specific inhibitors do not presently exist, however, there is a large number of inhibitors for the abundant NHE1 isoform, which is expressed on the plasma membrane of most cell types and which regulates cytosolic pH (*Masereel et al., 2003*). NHE1 inhibitors, such as the epithelial sodium channel blocker and guanidinium derivative amiloride, have been used in clinical practice for half a century as diuretics,

and numerous analogues have been developed over the years. We reasoned that some of these analogues might cross-inhibit NHE6 and thus decided to test them in our Apoer2 recycling assay. We found several amiloride analogues to be effective at restoring Apoer2 recycling and chose EMD87580 for detailed analysis (*Figure 4A*). Increasing concentrations progressively restored normal surface recycling of Apoer2 in the presence of ApoE4 (*Figure 4B,C*). As shown in *Figure 4D and E*, physiological concentrations of ApoE3 (5 μg/ml) also impaired Apoer2 recycling to a small, but significant extent. This was also prevented by EMD87580 at the same concentration (3 μM) at which the effect of ApoE4 on Apoer2 recycling was completely neutralized (*Figure 4D,E*), suggesting that ApoE3 also, although to a minor extent, may lose solubility upon entering the early endosome, and that this is also prevented by pH lowering. To confirm that the effect of the NHE inhibitor was caused by altering the pH and not by an unrelated mode of action we did the converse experiment. We incubated the neurons with ApoE4 in the presence of 3 μM EMD87580 and increasing concentrations of Bafilomycin, an inhibitor of the proton pump. *Figure 4F and G* show that the resulting increase of the endosomal pH reverses the correction of the ApoE4 recycling deficit by EMD87580 and results in renewed impairment of Apoer2 trafficking. This was further confirmed by an additional experiment in which we incubated the neurons in the presence of a fixed concentration of ApoE4 in the presence or absence of 50 nM Bafilomycin. In the presence of this partial inhibition of proton pump activity, higher concentrations of EMD87580 were required to restore normal Apoer2 recycling (*Figure 4H*) as evident by the right-shift of the dose-response curve (*Figure 4I*). Taken together, these data show that vesicular pH is the primary driving factor that determines to what extent ApoE4 alters endosomal trafficking.

## shRNA knockdown of NHE6 is sufficient to restore normal Apoer2 trafficking

More than 10 different NHEs exist in mammals (*Fuster and Alexander, 2014*), which function in intracellular, organellar and extracellular pH regulation in all cells of the body and in a variety of organs. EMD87580, as most other commercially available NHE inhibitors, was developed with the goal of inhibiting NHE1 (*Chen et al., 2004*) and its ability to cross inhibit other NHE forms is unknown. Because ApoE4 blocks Apoer2 and glutamate receptor recycling, and the early endosome

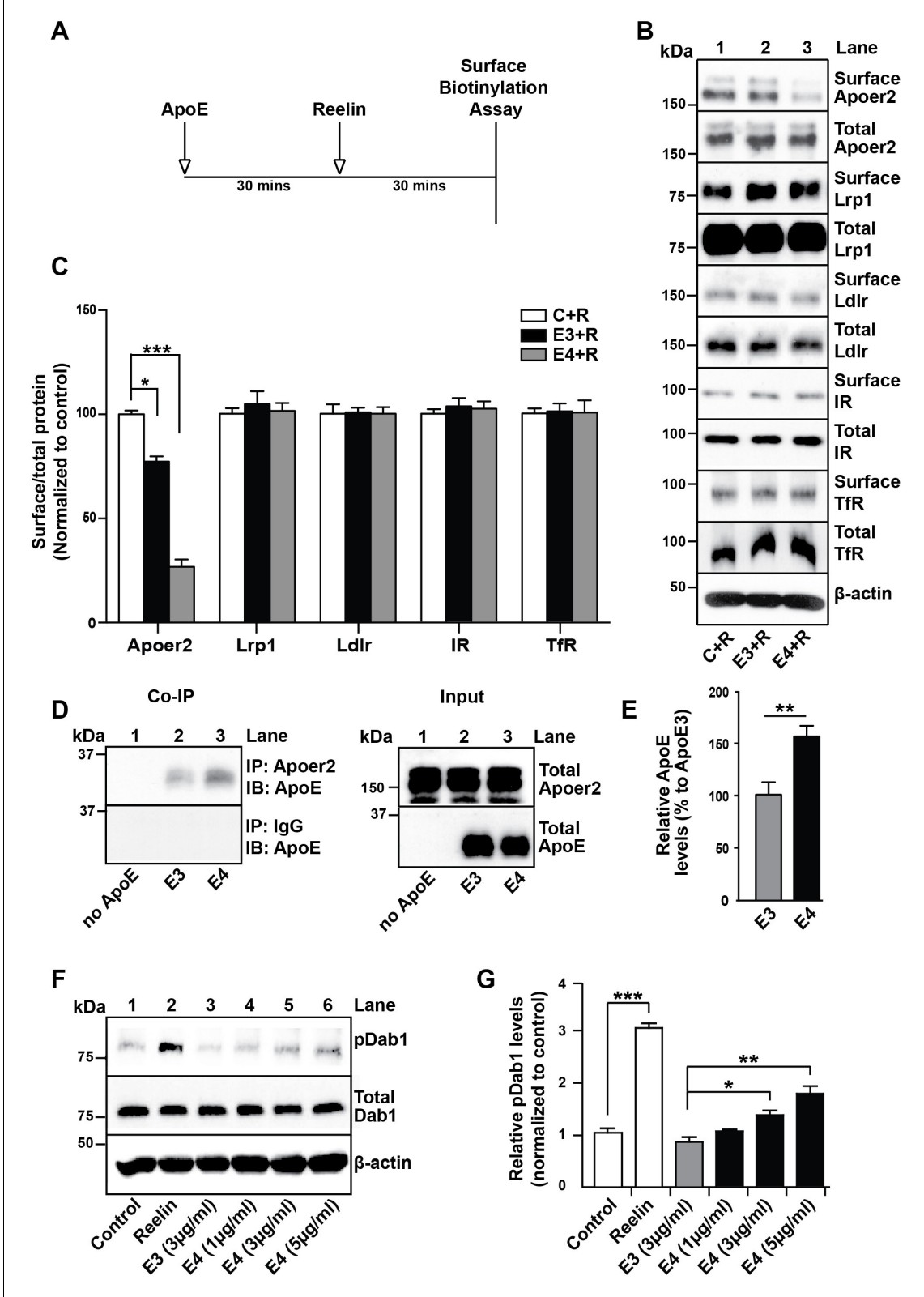

**Figure 2.** ApoE4 Impairs Recycling of the Reelin Receptor Apoer2. (**A**) Timeline for experiment shown in B and C. (**B and C**) Apolipoprotein E (ApoE) isoforms reduce surface expression of Apoer2. ApoE-conditioned media treatment reduces the surface expression of Apoer2 in presence of Reelin in primary neurons. Apoer2 surface levels show a higher reduction with ApoE4 than ApoE3. Other ApoE receptors, such as low-density lipoprotein receptor-related protein 1 (Lrp1) and low-density lipoprotein receptor (Ldlr), as well as the endocytic receptor for transferrin (TfR) and insulin receptor

*Figure 2 continued on next page*

*Figure 2 continued*

(IR) exhibit comparable surface levels in the presence of ApoE3 or ApoE4. Levels of surface proteins and total proteins were analyzed by immunoblotting using antibodies raised against Apoer2, Lrp1, Ldlr, IR and TfR. Quantitative analysis of the ratio of surface and total receptor levels is shown (C). (D and E) Proteins from primary neurons incubated with ApoE-conditioned media were immunoprecipitated with anti-Apoer2 or control rabbit IgG and immunoblotted with anti-ApoE antibody. Input is shown in the right panel of (D) and quantification in (E). (F and G) ApoE4, but not ApoE3, induces phosphorylation of Dab1 independent of Reelin. Primary neurons were incubated with ApoE-conditioned media or Reelin and tested for phospho-Dab1 and total Dab1. Quantitative analysis is shown (G). All data are expressed as mean ± SEM from three independent experiments. *p<0.05, **p<0.01, ***p<0.001. Statistical analysis was performed using one-way ANOVA and Dunnett's post-hoc test (C and G) or Student's *t*-test (E).
DOI: https://doi.org/10.7554/eLife.40048.007

The following source data is available for figure 2:

**Source data 1** ApoE4 Impairs Recycling of the Reelin Receptor Apoer2.
DOI: https://doi.org/10.7554/eLife.40048.008

is the first acidic organelle ApoE4 encounters during endocytosis, we suspected that the effect of EMD87580 to restore normal receptor trafficking is due to NHE6 inhibition. However, we could not exclude that other vesicular NHEs, in particular the NHE9, which resides in the Golgi and the late endosome, participate in the release of the recycling block. To determine whether NHE6 inhibition is sufficient to restore normal Apoer2 recycling, we used shRNAs designed against all the vesicular NHEs present in intracellular compartments which ApoE and Apoer2 might encounter during their passage through the recycling pathway. *Figure 5A–D* shows that of all NHEs (1 and 5 through 9) that were targeted by shRNA inhibition, only NHE6-specific shRNAs were able to completely restore Apoer2 expression at the plasma membrane in the presence of ApoE (*Figure 5A*, lane 6, 5C, lanes 8, 10, 12, and quantified in Panels B and D). Three different shRNAs directed against NHE6 were used in *Figure 5C and D*. None of the shRNAs directed against NHE1, 5, 7, 8 or 9 had any effect on surface Apoer2 expression, neither in the absence of ApoE4 or in its presence.

## shRNA knockdown of NHE6 restores normal trafficking of Apoer2, AMPA and NMDA receptors in the presence of ApoE4

Next, we determined whether shRNA-mediated inhibition of NHE6 function would also restore the normal recycling of AMPA and NMDA-type glutamate receptors in primary neurons treated with ApoE4 and Reelin. Three different shRNAs which efficiently reduced NHE6 protein expression by at least 90% (*Figure 6A*) and a scrambled control shRNA were used in the receptor surface recycling assay (*Figure 6B*, lanes 4 – 8). Pharmacological inhibition of NHE function with EMD87580 was used as a control (lane 3). EMD87580 and the NHE6-specific shRNAs completely restored Apoer2 and glutamate receptor surface expression, while the scrambled shRNA had no effect (Panel B and quantified in Panels C-F). These findings suggested that NHE6 inhibition might be effective in restoring a normal synaptic response in ApoE4 targeted replacement mice, which we have previously shown are completely resistant to long-term potentiation (LTP) enhancement by Reelin (*Chen et al., 2010*; *Lane-Donovan et al., 2014*).

## Pharmacological NHE inhibition restores Reelin-enhanced LTP in the presence of ApoE4

To test if restored glutamate receptor trafficking by NHE6 inhibition improves synapse function, we measured hippocampal LTP in acute slices of human ApoE targeted replacement mice: ApoE3-knockin (ApoE3-KI) and ApoE4-knockin (ApoE4-KI). Mice were treated with or without EMD87580 by simultaneous intraperitoneal and intranasal application, the brains were subsequently harvested and electrophysiological field recordings were performed. We chose to complement the intraperitoneal injections with intranasal delivery since EMD87580, as all existing guanidine-based NHEs, have poor blood-brain-barrier penetration and intranasal delivery of small molecules and peptides including insulin has been shown to increase their biological effect in the brain (*Grassin-Delyle et al., 2012*). Consistent with previous results (*Rönicke et al., 2009*), we found that EMD87580 increased input-output (I/O) ratios in the ApoE3-KI mice (*Figure 7A*). In the ApoE4-KI mice, baseline I/O ratios were higher and did not respond to EMD87580 (*Figure 7B*). As shown previously (*Durakoglugil et al., 2009*), LTP was increased in ApoE3-KI slices treated with Reelin (*Figure 7C*).

## A
### Distinguishing features of human ApoE

|        | 112 | 158 | relative charge | IEP |
|--------|-----|-----|-----------------|-----|
| ApoE2  | Cys | Cys | 0               | 5.9 |
| ApoE3  | Cys | Arg | +1              | 6.1 |
| ApoE4  | Arg | Arg | +2              | 6.4 |

**Figure 3.** Working model illustrating the hypthetical mechanism of the vesicular trafficking defect incurred by human Apoe4. (A) Cysteines/arginines at residues 112 and 158 account for the difference in relative charge and isoelectric point (IEP) of human ApoE isoforms. (B) Endosomal ApoE4/Apoer2 aggregates form upon acidification.

*Figure 3 continued on next page*

*Figure 3 continued*

Endosomal pH is regulated by the vacuolar-type H+-ATPase (vATPase, proton pump) and organellar Na+/H
+ exchangers (NHEs, proton leak). After binding to ApoE4, Apoer2 undergoes endocytosis, is sequestered in
endosomes and recycling is delayed. (**C**) Endosomal ApoE4/Apoer2 resolve when the pH is lowered further.
Accelerated acidification through NHE6 inhibition activity promotes dissociation of ApoE4 and Apoer2, resulting
in the efficient recycling of Apoer2 back to cell plasma membrane.
DOI: https://doi.org/10.7554/eLife.40048.009

ApoE3-KI slices treated with EMD87580 also showed increased LTP (*Figure 7C*). Interestingly, Reelin
and EMD87580 have no additional synergistic effect and in fact increase LTP to a lesser extent than
either EMD or Reelin alone. As shown previously (*Durakoglugil et al., 2009*), Reelin had no effect
on LTP in ApoE4-KI slices (*Figure 7D*). In contrast to ApoE3-KI, ApoE4-KI slices treated with
EMD87580 exhibited reduced LTP. Importantly, ApoE4-KI slices with EMD87580 readily responded
to Reelin, and LTP was increased. Therefore, on the ApoE4 knockin background, EMD87580 restores
electrophysiological parameters comparable to the ApoE3 and wild-type background.

## NHE inhibition counteracts Aβ-induced LTP suppression in ApoE4-KI mice

These data suggest that endosomal NHE inhibition can neutralize the effect of ApoE4 on vesicle traf-
ficking and concomitant synaptic dysfunction. Can it also reverse the persistent synaptic suppression
caused by oligomeric β-amyloid in the presence of ApoE4? Aβ$_{42}$ oligomers potently suppress synap-
tic potentiation (*Townsend et al., 2006*), but this can be averted by preincubation of hippocampal
slices with Reelin, which can by itself potentiate the synapse and thus counteract the Aβ induced
suppression (*Durakoglugil et al., 2009*). In *Figure 8*, we repeated these experiments again in the
absence as well as in the presence of EMD87580. As we had found previously, AD patient brain
extracts containing Aβ oligomers, but not control brain extracts, potently suppressed LTP in hippo-
campal slices from ApoE3-KI and in ApoE4-KI mice. Reelin prevented this suppression in the ApoE3
slices, while the slices from ApoE4 mice were almost completely resistant to Reelin and LTP
remained suppressed (*Figure 8*, solid triangles in Panels A and B). By striking contrast, this LTP sup-
pression in the presence of Aβ and Reelin in ApoE4 slices was completely abolished when the slices
were perfused with EMD87580 for 4 hr prior to LTP induction. In data not shown here we observed
at 30-min preperfusion with EMD87580 a trend toward alleviating the ApoE4-mediated Reelin resis-
tance that was, however, not yet significant. This may suggest that relief of the ApoE4 endosomal
recycling block requires some time, perhaps to 'flush out' the vesicles that are already stuck and
clog up the recycling route.

## Discussion

We have used insights into the molecular structure and biophysical properties of ApoE isoforms to
develop a novel rational drug identification approach to reverse the increased AD risk inherent to
the ApoE4 allele. In earlier studies, we found that ApoE4 impairs endosomal vesicle recycling
(*Chen et al., 2010*). While investigating the molecular basis for this trafficking delay, we recognized
that the predicted isoelectric point of ApoE4 closely matches the prevailing pH in the early endo-
some. We hypothesized that ApoE4, which is known to assume a molten-globule state under low pH
conditions, might lose solubility as it enters the lower pH environment of the early endosome. This in
turn could impair vesicle propagation through the endosomal recycling pathway and result in the
observed sequestration of ApoE receptors and associated excitatory neurotransmitter receptors in
cortical neurons (*Chen et al., 2010*). We predicted that changing endosomal pH away from the iso-
electric point of ApoE4 should prevent this isoelectric precipitation and resolve the recycling block.
Since raising endosomal pH using alkalinizing agents, such as ammonium chloride or chloroquine, is
known to arrest endosomal trafficking by preventing lipoprotein release from lipoprotein receptors
(*Goldstein et al., 1985*), we investigated approaches to lower endosomal pH instead. Two possible
mechanisms for this are i) activation of the proton pump, or ii) preventing proton efflux from the
endosome by inhibiting the endosomal sodium/potassium hydrogen exchanger NHE6. Our results
now show that pharmacological as well as genetic inhibition of NHE activity in the early endosome is

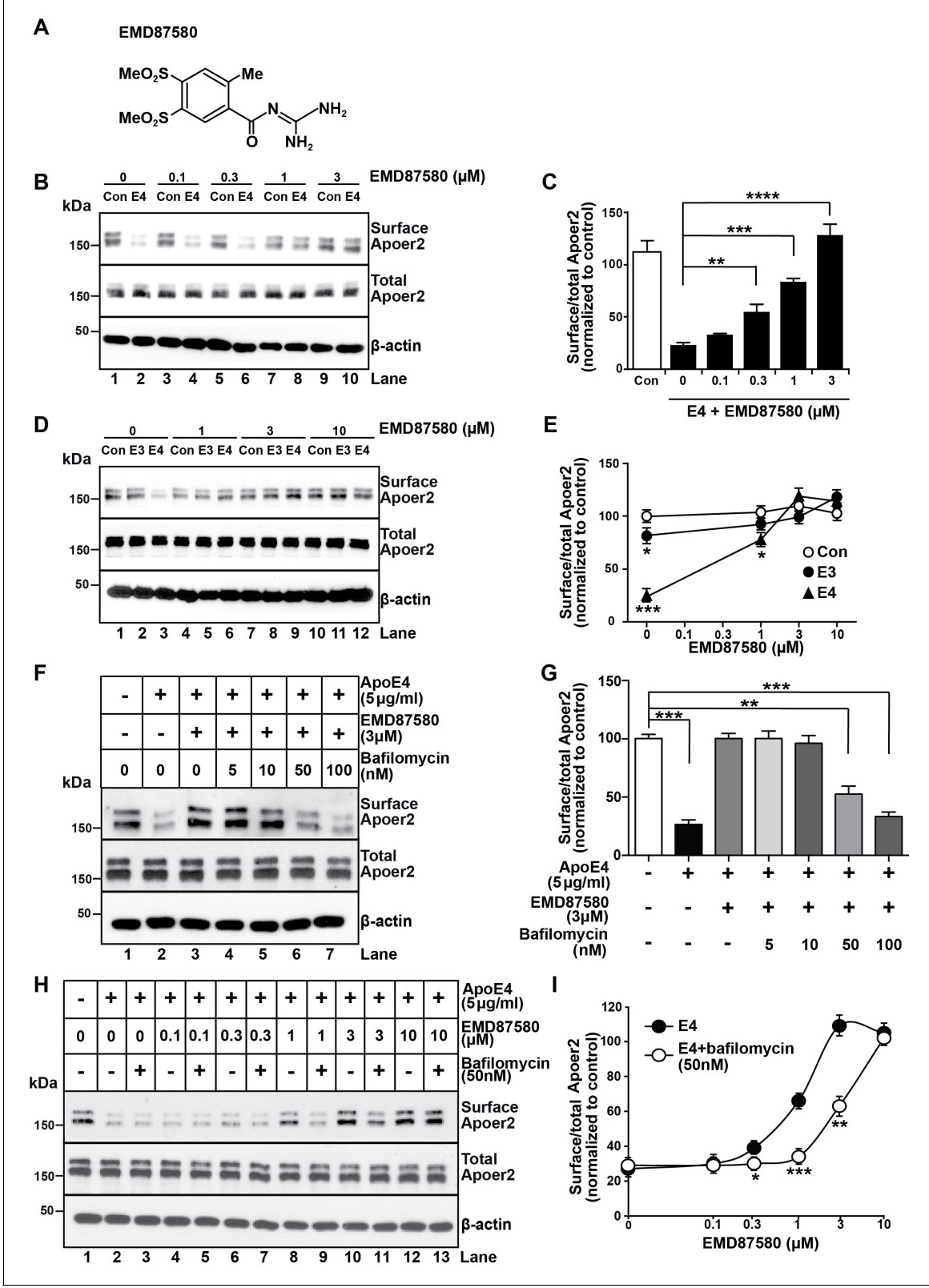

**Figure 4.** The NHE inhibitor EMD87580 prevents the intracellular trapping of ApoE4 and its receptor Apoer2. (**A**) Chemical structure of the NHE inhibitor EMD87580. (**B and C**) EMD87580 increases the Reelin-induced surface expression of Apoer2 in ApoE4-treated neurons in a dose-dependent manner. Primary neurons were pre-treated with EMD87580 at the indicated concentrations and then incubated with Reelin with or without cell-derived ApoE4. Surface and total Apoer2 levels were analyzed by immunoblotting. (**D and E**) The effect of EMD87580 on Reelin-induced Apoer2 trafficking in

*Figure 4 continued on next page*

*Figure 4 continued*

the presence of ApoE3 or ApoE4. Primary neuronal cells were treated with EMD87580, Reelin and either ApoE3- or ApoE4-conditioned media. (**F and G**) Bafilomycin, a proton pump inhibitor, counteracts the effect of EMD87580 on Apoer2 recycling in a dose-dependent manner. Primary neurons were pre-treated with or without bafilomycin in the presence or absence of EMD87580 and subsequently incubated with ApoE4 and Reelin. Surface and total Apoer2 levels were analyzed by immunoblotting. (**H and I**) Bafilomycin shifts the EMD87580 dose response curve of Apoer2 surface expression. All values are expressed as mean ±SEM from three independent experiments. *$p < 0.05$, **$p < 0.01$, ***$p < 0.001$. Statistical analysis was performed using one-way ANOVA and Dunnett's post-hoc test (**C, E and G**) or Student's *t*-test (**I**).

DOI: https://doi.org/10.7554/eLife.40048.010

The following source data is available for figure 4:

**Source data 1.** The NHE inhibitor EMD87580 prevents the intracellular trapping of ApoE4 and its receptor Apoer2.
DOI: https://doi.org/10.7554/eLife.40048.011

sufficient to completely resolve the ApoE4 induced endosomal recycling block and restore the normal cell surface recycling rate of the synaptic ApoE receptor Apoer2 and the excitatory AMPA- and NMDA-type glutamate receptors that are regulated by Apoer2 and that traffic together with Apoer2 through the endosomal recycling compartments (illustrated by the model shown in *Figure 9*).

Previously, we and others have shown that ApoE4-KI mice exhibit enhanced LTP while the neuromodulator Reelin has no potentiating effect on LTP expression in this genotype (*Durakoglugil et al., 2009*), suggesting that ApoE4 impairs the physiological response to Reelin. Here, we show that NHE inhibition with EMD87580 in ApoE4-KI mice reverses the ApoE4 induced Reelin resistance (*Figure 7*) and restores the ability of Reelin to protect against Aβ toxicity (*Figure 8*). ApoE3-KI slices treated with EMD87580 exhibit increased LTP consistent with previous findings that have shown that the NHE inhibitor ethyl-isopropyl amiloride (EIPA) can enhance theta-burst-induced LTP (*Rönicke et al., 2009*). NHE activity was proposed to be a negative feedback mechanism that can regulate neuronal excitability as well as plasticity. In ApoE4-KI mice, EMD87580 decreased LTP to baseline levels of control ApoE3-KI mice and importantly LTP became now responsive to Reelin-facilitation. This observation suggests that it is the restoration of normal release of ApoE4 from Apoer2 in acidified endosomal compartments and the subsequent normalization of endosomal trafficking that reestablishes optimal synaptic homeostasis in the presence of ApoE4.

Although more than 20 genetic loci that modify the risk for late-onset AD have been discovered to date (*Karch et al., 2014*), ApoE4 genotype is by far the major genetic risk factor for late-onset AD besides aging, affecting almost 1/5[th] of the human population, and hence it is clinically the most important one. The molecular mechanisms by which ApoE4 imposes this risk remain under debate. Early work following the seminal discovery of this striking genetic association by the Roses group (*Corder et al., 1993*) focused on the differential ability of ApoE isoforms to interact with Aβ and affect fibril formation. Efforts in our own laboratory were based on the rationale that ApoE receptors, that is the LDL receptor gene family, are highly likely to be involved in the disease process. This hypothesis was the initial driver that defined a plethora of surprising functions of LDL receptor-related proteins (LRPs) in the central and peripheral nervous system (*Bal et al., 2013*; *Beffert et al., 2005*; *Bell et al., 2012*; *Choi et al., 2013*; *Kim et al., 2008*; *Lane-Donovan and Herz, 2017*; *Liu et al., 2013*; *Liu et al., 2007*; *Liu et al., 2011*; *May et al., 2004*; *Nakajima et al., 2013*; *Pohlkamp et al., 2015*; *Pohlkamp et al., 2017*; *Trommsdorff et al., 1999*; *Wasser and Herz, 2017*; *Wasser et al., 2014*; *Weeber et al., 2002*; *Zhang et al., 2008*; *Zhao et al., 2017*). They included unprecedented roles as direct signal-transducing receptors (*Hiesberger et al., 1999*; *Trommsdorff et al., 1999*) and regulators of central and peripheral synaptic transmission (*Beffert et al., 2005*; *Choi et al., 2013*; *Weeber et al., 2002*) and have established a strong rationale and mechanistic basis by which ApoE isoforms and ApoE receptors can directly affect synaptic homeostasis, neuronal survival and thus the cognitive impairment and progressive neurodegeneration that underlie LOAD.

Presynaptic and postsynaptic vesicle recycling is a central element of synaptic transmission (*Harris et al., 2012*; *Kawasaki et al., 2000*; *Robinson et al., 1993*; *Sontag et al., 1994*; *Sudhof, 2004*). Intriguingly, enlarged endosomal compartments and impaired endolysosomal functions are also a prominent feature of APP expression, processing and early AD (*Cataldo et al., 2000*; *Decourt et al., 2013*; *Ishigaki et al., 2000*; *Nixon et al., 2001*; *Salehi et al., 2006*). Our original finding that ApoE isoforms can differentially impair synapse functions by trapping postsynaptic (and

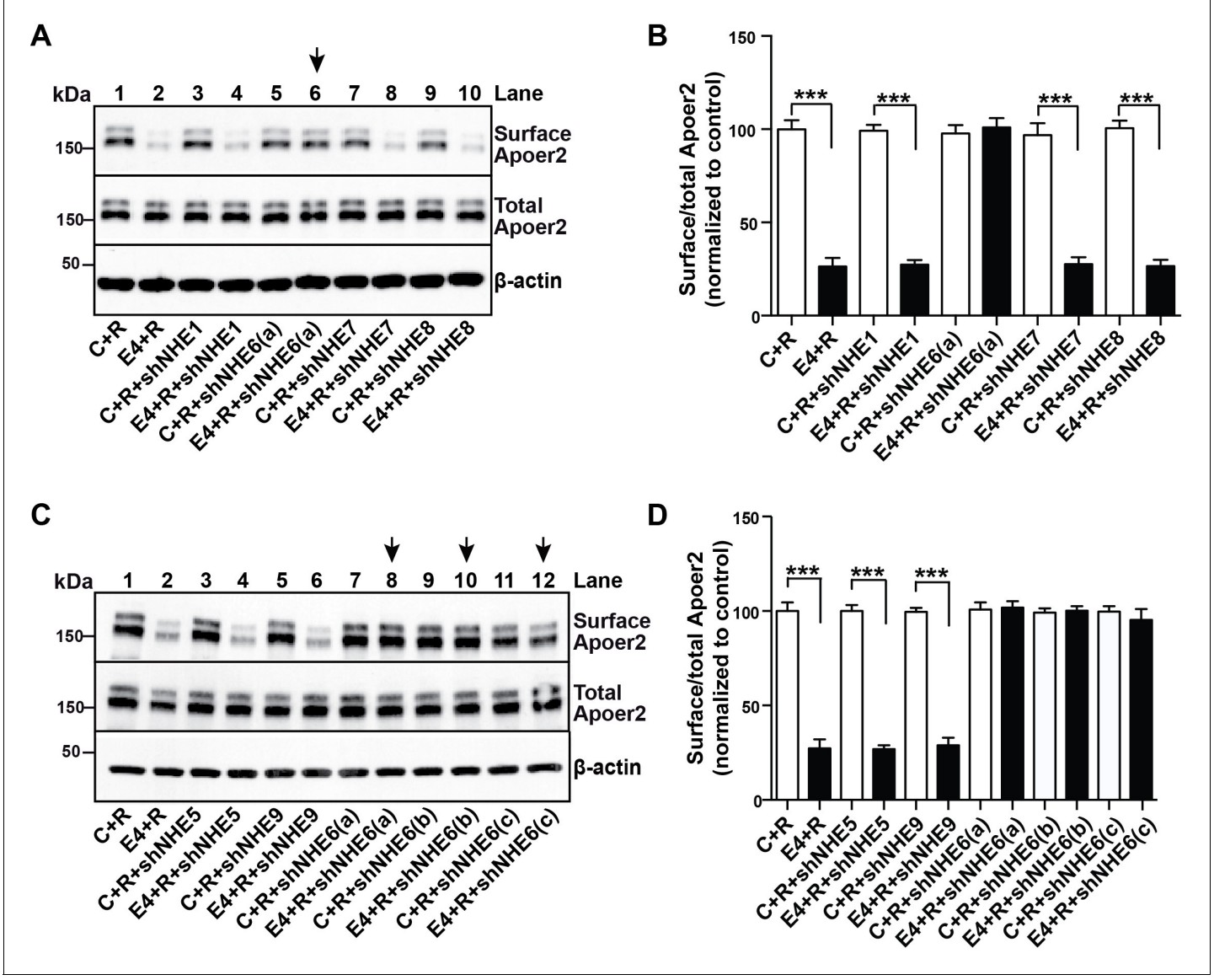

**Figure 5.** A Specific Role for NHE6 in Apoer2 trafficking. (**A and C**) shRNA knockdown of NHE6, but not other NHEs (NHE1, 5, 7, 8, 9) restores ApoE4-impaired Apoer2 recycling. Lentivirus-mediated shRNAs targeting NHE1, 5, 6, 7, 8, or 9 were applied to primary neurons. Cells were then treated with ApoE4-conditioned media and Reelin, and cell surface and total Apoer2 were determined by immunoblotting. Arrows indicate conditions with restored Apoer2 surface levels. Three different shRNA constructs against NHE6 showed significant attenuation of Apoer2 cell surface levels (shNHE6 a, b, c). (**B and D**) Quantitative analysis of (**A**) and (**C**). All data are expressed as mean ±SEM of three independent experiments. ***p<0.001. Statistical analysis was performed using Student's *t*-test (**B** and **D**).

DOI: https://doi.org/10.7554/eLife.40048.012

The following source data is available for figure 5:

**Source data 1.** A Specific Role for NHE6 in Apoer2 trafficking.

DOI: https://doi.org/10.7554/eLife.40048.013

possibly also presynaptic) recycling vesicles that contain ApoE receptors was inspired by the original observations of Heeren, Beisiegel and colleagues who first described a prolonged intracellular retention of ApoE4 in a hepatoma cell line (*Heeren et al., 2004*). In a series of experiments, we showed that ApoE4 impairs NMDA receptor activation and neuronal $Ca^{2+}$ conductance by the neuromodulator and ApoE receptor ligand Reelin. This was caused by the dramatically delayed recycling of the Reelin receptor and regulator of glutamate receptor trafficking Apoer2 in the presence of ApoE4 and, notably, also to a smaller extent by ApoE3 and less by ApoE2 (*Chen et al., 2010*). The

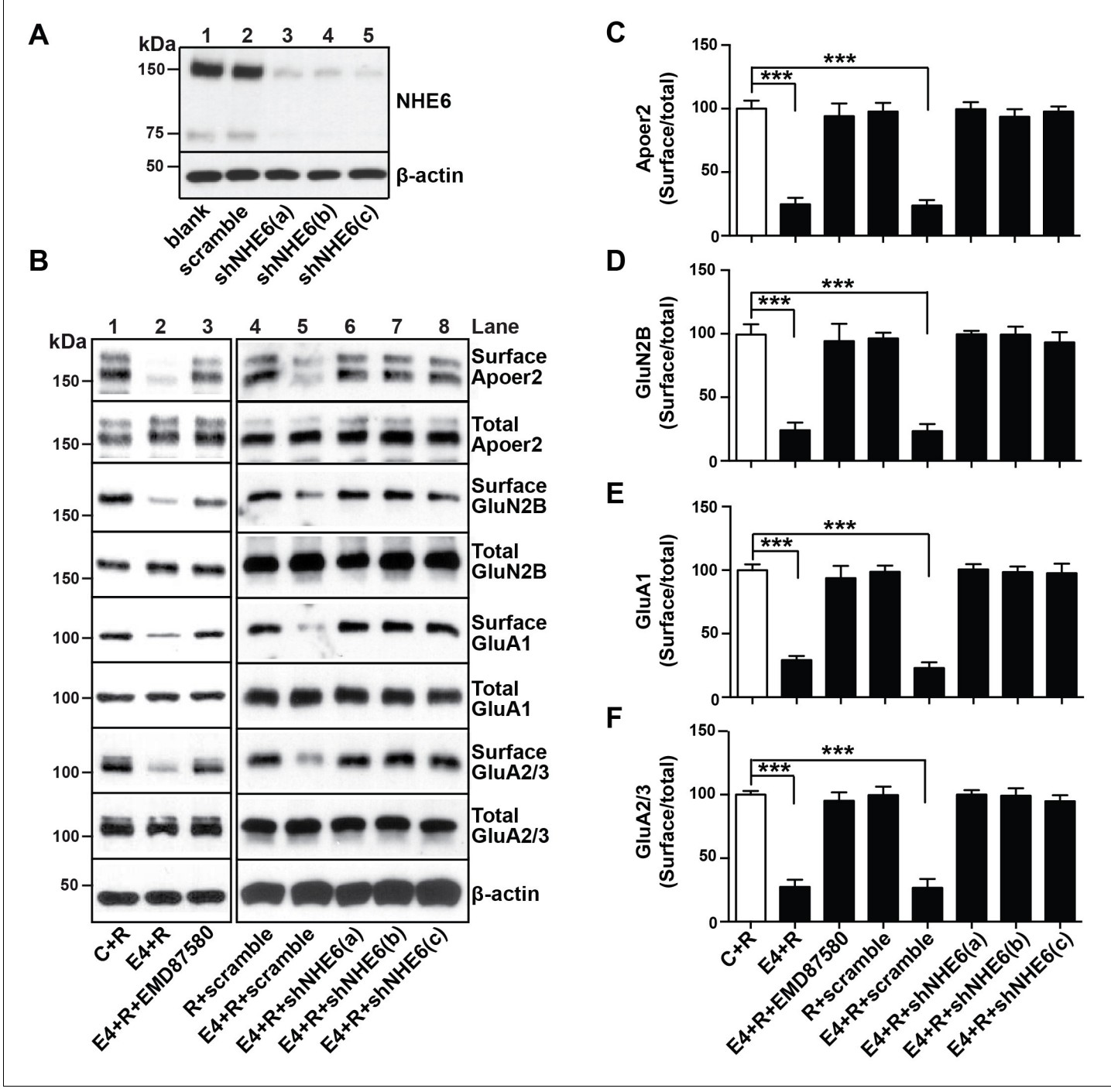

**Figure 6.** NHE6 knockdown alleviates surface trafficking deficits induced by ApoE4. (**A**) Lentiviral shRNA knockdown efficiency of NHE6 protein expression in primary rat cortical neurons. (**B**) Lentiviral shRNA directed against NHE6 restores the ApoE4-induced trafficking deficits of surface receptors. Primary cortical neurons were infected with three different lentiviral shRNAs directed against NHE6 (shNHE6 a, b and c; lanes 6–7) or scrambled shRNA control (lanes 4, 5). Infected cultures were treated without (lanes 1 and 4) or with (lanes 2, 3, 5–8) cell-derived ApoE4 and Reelin (all lanes) and the cell surface biotinylation assay was performed for Apoer2, GluN2B, GluA1 and GluA2/3. (**D–F**) Quantitative analysis of immunoblot signal from (**B**). All data are expressed as mean ±SEM from three independent experiments. *p<0.05, **p<0.01, ***p<0.001. Statistical analysis was performed using one-way ANOVA and Dunnett's post-hoc test (**C–F**).

DOI: https://doi.org/10.7554/eLife.40048.014

The following source data is available for figure 6:

**Source data 1.** NHE6 knockdown alleviates surface trafficking deficits induced by ApoE4.

DOI: https://doi.org/10.7554/eLife.40048.015

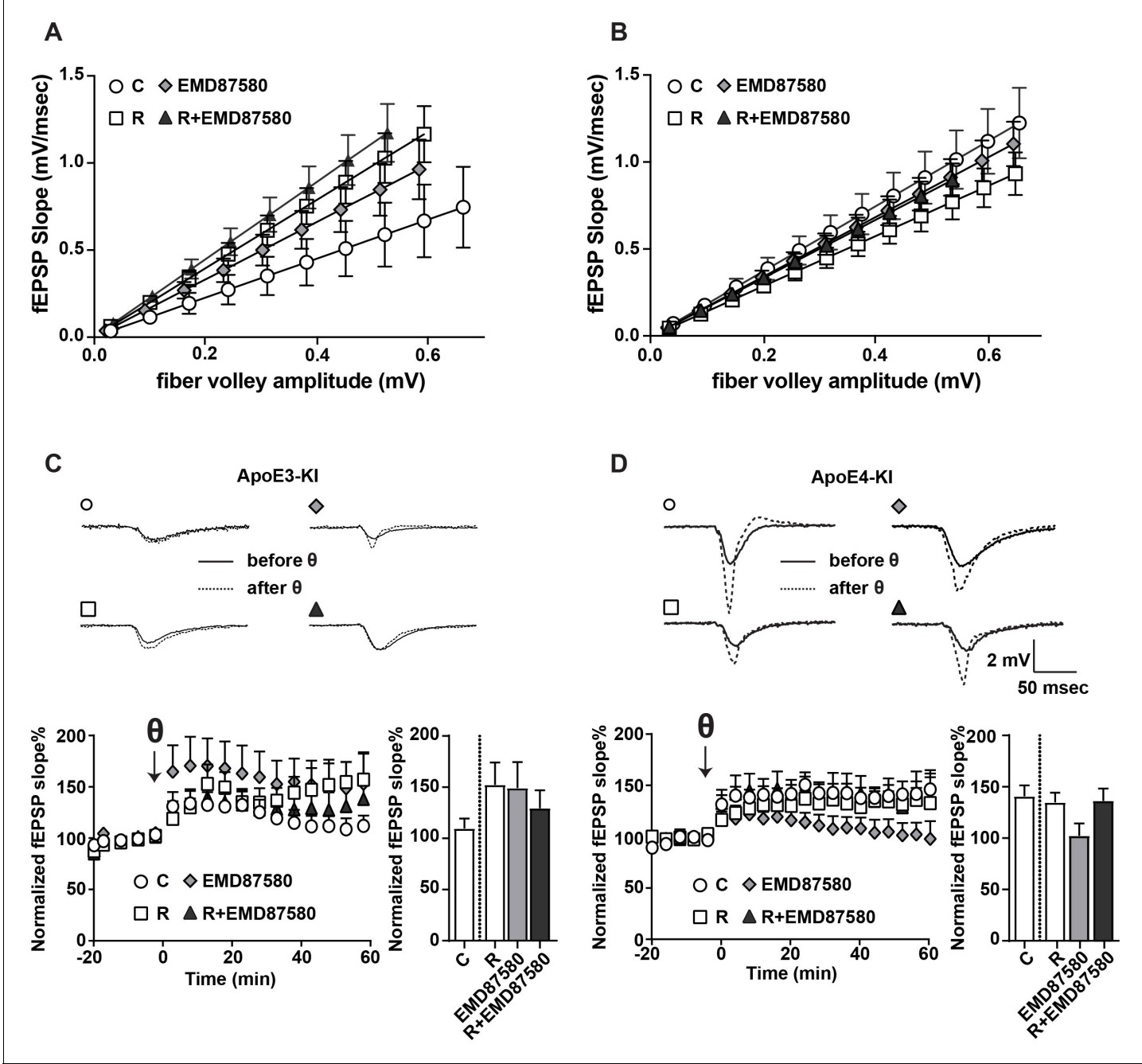

**Figure 7.** EMD87580 treatment differentially alters synaptic plasticity in ApoE3-KI and ApoE4-KI mice. Mice were pre-treated with EMD87580 in vivo and acute hippocampal slices were subsequently analyzed by recording extracellular field potentials (**A and B**) Input-output curves are shown for ApoE3-KI (**A**) and ApoE4-KI (**B**). (**A**) ApoE3-KI slices treated with Reelin or treated with EMD87580 and Reelin exhibited increased I/O slopes compared to control (Ctrl: 1.127 ± 0.18; EMD87580: 1.653 ± 0.15; Reelin: 1.97 ± 0.14; Reelin and EMD87580: 2.23 ± 0.16; F = 9.567, p<0.05). (**B**) I/O curves were increased in ApoE4-KI slices at baseline (1.86 ± 0.16) compared to ApoE3-KI control (1.127 ± 0.18). Neither EMD87580 nor Reelin significantly affected the I/O slopes in ApoE4-KI slices (EMD: 1.705 ± 0.10; Reelin: 1.43 ± 0.09; Reelin and EMD: 1.67 ± 0.09; F = 1.8, p=0.14). (**C and D**) Results from LTP recordings in ApoE3-KI (**C**) and ApoE4-KI (**D**). Representative traces before (solid line) and 40 min after (dashed line) theta-burst stimulation (TBS) for each treatment paradigm are shown in the top panels. Bottom panels depict LTP recordings and quantification of average LTP responses between 40 and 60 min after TBS (bar graphs). (**C**) ApoE3-KI slices treated with Reelin (152.4% ± 21.69, n = 5) and EMD87580 (149.30 ± 25.29, n = 7) had increased LTP compared to control slices (109.7% ± 9.7, n = 6). Combined Reelin treatment with EMD87580 increased LTP (129.4 ± 17.6, n = 7) compared to control. (**D**) Untreated ApoE4-KI slices showed enhanced LTP (140.80% ± 10.5, n = 12) when compared to untreated ApoE3-KI slices (109.7% ± 9.7, n = 6). ApoE4-KI slices treated with Reelin did not further potentiate LTP (134.70% ± 9.63, n = 14), whereas ApoE4-KI slices treated with EMD87580 exhibited reduced LTP (102 ± 11.9, n = 14). ApoE4-KI slices with EMD87580 show increased LTP when treated with Reelin (136.30 ± 11.87, n = 15) as

*Figure 7 continued on next page*

*Figure 7 continued*

compared to EMD87580 treatment alone. Open circles: no additions; Open squares: Reelin alone; Gray diamonds: EMD87580 alone; Filled triangles: Reelin and EMD87580 treated.

DOI: https://doi.org/10.7554/eLife.40048.016

The following source data is available for figure 7:

**Source data 1.** EMD87580 treatment differentially alters synaptic plasticity in ApoE3-KI and ApoE4-KI mice.

DOI: https://doi.org/10.7554/eLife.40048.017

dramatically altered recycling kinetics in the presence of ApoE4 lead to an altered state of synaptic homeostasis, which we have termed 'Reelin resistance' (*Lane-Donovan et al., 2014*) and which prevents the synapse from adequately adapting to the rising levels of synapse-suppressing Aβ oligomers as they accumulate in the aging brain. The compensatory increase in synaptic and network activity (*Palop and Mucke, 2010*) would further drive Aβ production (*Cirrito et al., 2008*), thereby accelerating a self-reinforcing cycle that would be predicted to contribute to the earlier amyloid accumulation in ApoE4 carriers. Moreover, a second mechanism by which impaired ApoE4-containing vesicle recycling would be predicted to contribute to accelerated amyloid accumulation and plaque deposition (*Fagan et al., 2002*) involves the reduced turnover rate of Aβ in the brains of ApoE4 targeted replacement mice (*Castellano et al., 2011*) and humans (*Wildsmith et al., 2012*).

Recently, the Bu laboratory reported a direct interaction of ApoE with insulin receptors in the brain, which also resulted in their intracellular retention and impaired insulin signaling (*Zhao et al., 2017*). Although in our experiments we did not detect impaired insulin receptor trafficking (*Figure 2B and C*), and there is also no evidence that ApoE4 carriers are predisposed to the predicted insulin resistance that would be expected to result from intracellular insulin receptor trapping, these data add nevertheless further support to our model which postulates impaired neuronal endosomal vesicle trafficking as the root cause for the increased AD risk imposed by ApoE4 (*Chen et al., 2005*; *Chen et al., 2010*; *Lane-Donovan and Herz, 2017*; *Lane-Donovan et al., 2014*).

Our data indicate that ApoE4 induces endosomal trafficking deficits. Consistently, alkalinizing drugs, such as ammonium chloride or chloroquine, which increase vesicular pH in the cell, lead to an arrest of endosomal trafficking by preventing lipoprotein release from their receptors (*Goldstein et al., 1985*). We therefore chose to explore the effect of acidification of endosomes as a means to overcome the trafficking impairments caused by ApoE4. By contrast, a recent study in astrocytes has proposed that the presence of ApoE4 results in endosomal acidification caused by NHE6 reduction, leading to impaired Aβ-clearance (*Prasad and Rao, 2018*). These authors further reported that increased NHE6 expression, which would elevate endosomal pH, induced by HDAC-inhibition alleviated the impaired of Aβ-clearance.

We show here that Aβ-induced synaptic impairment could be abolished by the NHE inhibitor EMD87580 (*Figure 8*). Both studies show that manipulation of endosomal pH can affect the endosomal trafficking of ApoE. We show that in neurons this alters cell surface Apoer2 expression levels and thus synaptic plasticity. *Prasad and Rao (2018)* showed that in astrocytes a similar mechanism may mediate Aβ clearance involving the ApoE receptor Lrp1. The isoelectric point of ApoE4, but not ApoE2 and ApoE3, matches the physiological pH present in early endosomes and reduced solubility at or near their isoelectric point is a general property of many proteins including insulin (*Wintersteiner and Abramson, 1933*). In contrast to ApoE2 and ApoE3, the conformation of ApoE4 in the early endosome might be particularly vulnerable, because it alone has been shown to be prone to unfolding under low pH conditions resulting in a molten-globule state (*Morrow et al., 2002*).

Along the same lines, ApoE4 increased the number and size of early endosomes in AD patients (*Cataldo et al., 2000*) and in neurons derived from induced pluripotent stem cells (*Lin et al., 2018*). These data suggest that manipulating endosomal pH represents a promising therapeutic target to improve endosomal trafficking deficits induced by ApoE4.

In order to fully address the physiological functions of NHE6 and understand its role in ApoE4 induced neurodegenerative processes, it will be necessary to develop NHE6 specific inhibitors that can pharmacologically modulate NHE6 - as opposed to switching it on or off in a binary fashion - to achieve the optimal pH for ApoE4 trafficking without undue interference with essential cellular

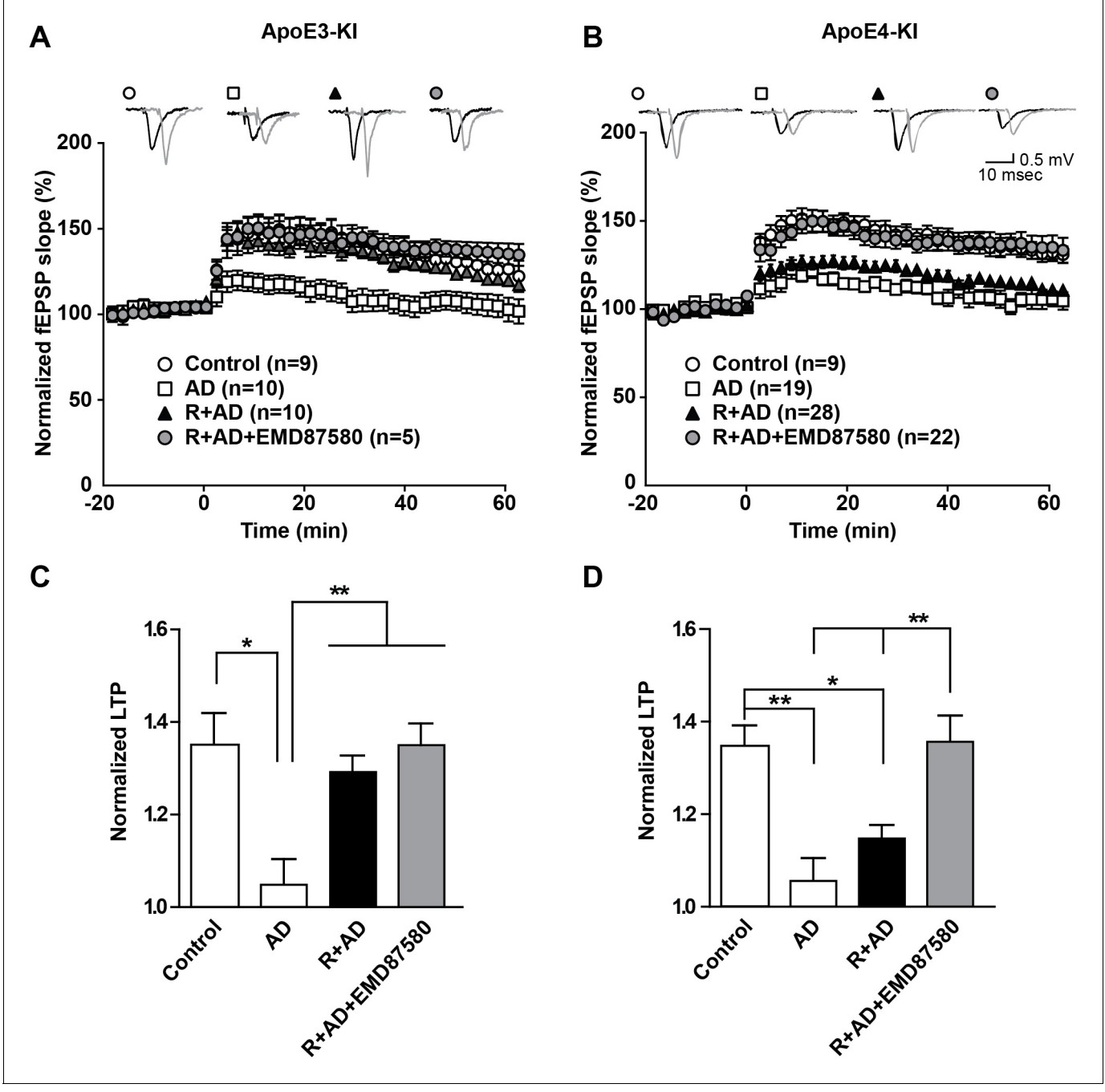

**Figure 8.** NHE inhibition counteracts Aβ-induced LTP suppression in ApoE4-KI mice. (A–D) Treatment of hippocampal slices with AD brain extracts impairs long-term potentiation (LTP) in ApoE3-KI and ApoE4-KI mice. Reelin can attenuate the LTP deficits induced by AD extracts in ApoE3-KI, but not ApoE4-KI mice. Inhibition of NHE counteracts the LTP deficits induced by AD extract in ApoE4-KI mice. Hippocampal slices were prepared from 2- to 3-month-old ApoE3-KI and ApoE4-KI mice. Extracellular field recordings were performed in slices treated with AD brain extract, Reelin and/or EMD87580. Control slices were treated with control brain extract. Theta burst stimulation (TBS) was performed 20 min after stable baseline was attained. Representative traces are shown in each panel, before TBS induction (black) and 40 min after TBS (grey). (C, D) Quantitative analysis of normalized fEPSP slopes at 40–60 min post TBS for (A), respectively (B). All data are expressed as mean ±SEM. *p<0.05, **p<0.01. Statistical analysis was performed using one-way ANOVA followed by Tukey's post-hoc test (C, D).

DOI: https://doi.org/10.7554/eLife.40048.018

The following source data is available for figure 8:

**Source data 1.** NHE inhibition counteracts Aβ-induced LTP suppression in ApoE4-KI mice.

DOI: https://doi.org/10.7554/eLife.40048.019

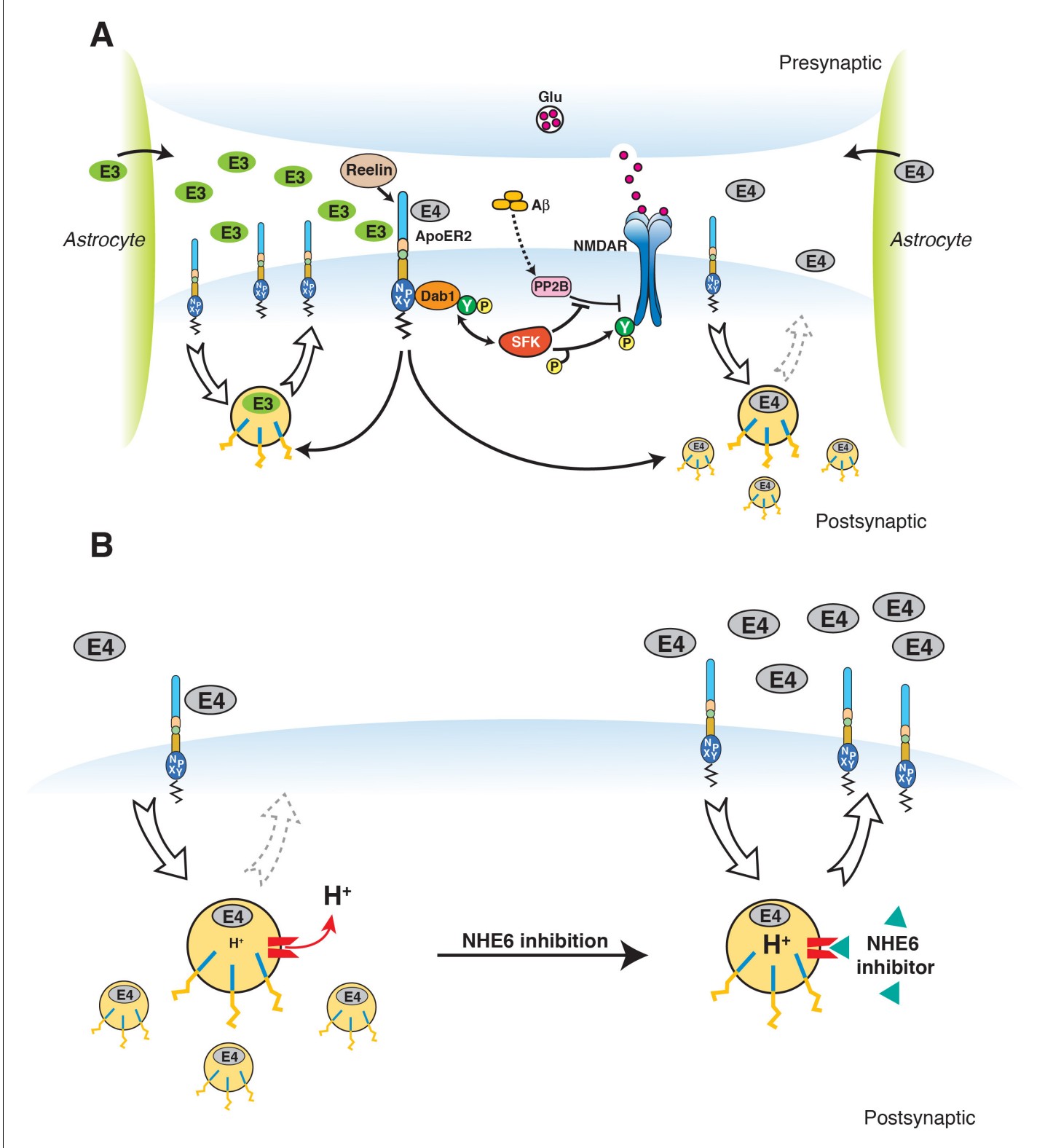

**Figure 9.** Restoration of vesicular trafficking and synaptic homeostasis by NHE6 inhibition in the presence of ApoE4. (**A**) Effect of ApoE isoforms on ApoE receptor signaling at the synapse. Apoer2 induces NMDAR tyrosine phosphorylation by activating SFKs in response to Reelin in the postsynaptic neuron. Astrocyte-derived ApoE3 (green ovals) or ApoE4 (gray ovals) bind to Apoer2 and are constitutively but slowly internalized. Apoer2 undergoes accelerated endocytosis in response to Reelin signaling. ApoE4 sequesters Apoer2 in intracellular compartments thereby reducing the ability of the postsynaptic neuron to phosphorylate (activate) NMDA receptors in response to Reelin (shown on the right), whereas ApoE2 or ApoE3 efficiently

*Figure 9 continued on next page*

*Figure 9 continued*

recycle back to the cell surface and thus deplete surface Apoer2 levels to a lesser extent (illustrated on the left for ApoE3). Aβ oligomers interfere with NMDAR tyrosine phosphorylation by activating tyrosine phosphatases. This panel has been reproduced from *Chen et al. (2010)*. (B) When NHE6 (red) is functioning normally, early endosomal pH is maintained at ~6.5 by the action of the proton pump and proton leakage through NHE6. This pH level is close to the isoelectric point (pI) of ApoE4, which is particularly sensitive to structural unfolding into a molten globule state upon entering acidic compartments (*Morrow et al., 2002*). The resulting reduced solubility may hamper efficient release of ApoE4 from its receptors (such as Apoer2 in neurons). Upon NHE6 inhibition, endosomal pH is further reduced, resulting in the release of ApoE4 from its receptors through improved solubility and the repulsion forces proteins exert upon each other when the pH is below their pI. Thus, trafficking and recycling of Apoer2 and the neurotransmitter receptors it associates with is restored. It should be noted that this property of ApoE4 to undergo impaired trafficking through endosomal compartments is universal and not restricted to neurons. Therefore, NHE6 inhibition may also aid in the clearance of amyloid-β in the brain and LDL removal in the liver.

DOI: https://doi.org/10.7554/eLife.40048.020

transport processes or the function of other NHEs. Such inhibitors must also be capable of crossing the blood brain barrier, something the presently available non-specific inhibitor classes are incapable of doing.

In summary, we have shown that the conformational change of ApoE4 to a molten-globule state (*Morrow et al., 2002*) in a low pH environment, which correlates with the impaired endosomal vesicle recycling in the presence of ApoE4 (*Chen et al., 2010*; *Heeren et al., 2004*; *Lane-Donovan and Herz, 2017*; *Lane-Donovan et al., 2014*; *Rellin et al., 2008*), can be reversed by changing endosomal pH (*Herz et al., 2010*). As a mechanistic basis we propose the propensity of many proteins to lose hydrophilicity at or near their isoelectric point, which makes them prone to aggregation and precipitation, a seminal discovery that made the purification of insulin on an industrial scale possible (*Wintersteiner and Abramson, 1933*). By altering endosomal pH, ApoE4 maintains its solubility and the recycling block is avoided. This simple biophysical property can explain in a straightforward manner many observations by which ApoE isoforms differentially affect neuronal functions and AD-relevant mechanisms. Our findings also point toward NHE6-specific inhibitors as a rational basis for a novel approach to erase the AD risk imposed by ApoE4.

# Materials and methods

**Key resources table**

| Reagent type (species) or resource | Designation | Source or reference | Identifiers | Additional information |
|---|---|---|---|---|
| Cell line (*Homo sapiens*) | HEK293 | Thermo Fisher | R70507, RRID: CVCL_0045 | Tested mycoplasma free annually, last test January 16,2018 |
| Cell line (*Homo sapiens*) | HEK293T | ATCC | CRL-3216, RRID: CVCL_0063 | Tested mycoplasma free annually, last test January 16,2018 |
| Strain, strain background (*Mus musculus*) | Mouse/ApoE3ki (B6.129P2- Apoetm2 (APOE*3)Mae N8) | (*Knouff et al., 1999*; *Sullivan et al., 1997*) | Originally provided by Dr. Nobuyo Maeda | |
| Strain, strain background (*Mus musculus*) | Mouse/ApoE4ki (B6.129P2- Apoetm2 (APOE*4)Mae N8) | (*Knouff et al., 1999*; *Sullivan et al., 1997*) | Originally provided by Dr. Nobuyo Maeda | |
| Strain, strain background (*Rattus norvegicus*) | SD rat | Charles River | SC:400 | |

*Continued on next page*

*Continued*

| Reagent type (species) or resource | Designation | Source or reference | Identifiers | Additional information |
|---|---|---|---|---|
| Antibody | goat anti-ApoE, pAb | EMD Millipore | 178479, RRID: AB_10682965 | 1:1000 (WB) |
| Antibody | rabbit anti-Apoer2 | Herz Lab, #2561 | | 1:1000 (WB) |
| Antibody | rabbit anti-Dab1 | Herz Lab, #5091 | | 1:1000 (WB) |
| Antibody | mouse anti-FLAG M2 | Sigma-Aldrich | F3165, RRID: AB_259529 | 1:1000 (WB) |
| Antibody | rabbit anti-GluA1 | Abcam | ab31232, RRID: AB_2113447 | 1:1000 (WB) |
| Antibody | rabbit anti-GluA2/3 | EMD Millipore | 07–598, RRID: AB_310741 | 1:1000 (WB) |
| Antibody | rabbit anti-GluN2B | Cell Signaling Technology | 4207S, RRID: AB_1264223 | 1:1000 (WB) |
| Antibody | rabbit anti-Insulin Receptor B (4B8), mAb | Cell Signaling Technology | 3025, RRID: AB_2280448 | 1:1000 (WB) |
| Antibody | rabbit anti-Lrp1 | Herz Lab | | 1:5000 (WB) |
| Antibody | rabbit anti-Ldlr | Herz Lab | | 1:1000 (WB) |
| Antibody | rabbit anti-NHE6 (C-terminus) | Herz Lab | | 1:1000 (WB) |
| Antibody | mouse anti-phosphotyrosine (4G10) mAb | EMD Millipore | 05–321, RRID: AB_309678 | 1:1000 (WB) |
| Antibody | rabbit anti-Transferrin receptor | Abcam | Ab61134, RRID: AB_943620 | 1:1000 (WB) |
| Antibody | rabbit anti-B-Actin | Abcam | Ab8227, RRID: AB_2305186 | 1:3000 (WB) |
| Peptide, recombinant protein | 6-cyano-7-nitroquinoxaline-2,3-dione, CNQX | Sigma-Aldrich | C127 | |
| Peptide, recombinant protein | ApoE3, human | Sigma-Aldrich | SRP4696 | |
| Chemical compound, drug | B-27 Supplement (50X), serum free | Thermo Fisher | 17504044 | |
| Chemical compound, drug | Bafilomycin A1 | Cayman Chemical | CAS88899-55-2 | |
| Chemical compound, drug | DMEM | Sigma-Aldrich | D6046 | |
| Chemical compound, drug | FuGENE | Promega | E2311 | |
| Chemical compound, drug | HBSS (1X) | Gibco | 14175 | |
| Chemical compound, drug | L-Glutamic acid (Glutamate) | Sigma-Aldrich | G1251 | |
| Chemical compound, drug | Neurobasal Medium (1X) Liquid without Phenol Red | Thermo Fisher | 12348017 | |

*Continued on next page*

Continued

| Reagent type (species) or resource | Designation | Source or reference | Identifiers | Additional information |
|---|---|---|---|---|
| Chemical compound, drug | NeutrAvidin Agarose | Thermo Fisher | 29201 | |
| Chemical compound, drug | Nimodipine | Sigma-Aldrich | N3764 | |
| Chemical compound, drug | NP-40 Alternative | EMD Millipore | 492016 | |
| Chemical compound, drug | 32% Paraformaldehyde AQ solution | Fisher Scientific | 15714S | |
| Chemical compound, drug | PBS (1X) | Sigma-Aldrich | D8537 | |
| Chemical compound, drug | Penicillin-Streptomycin Solution, 100X | Corning | 30–002 CI | |
| Chemical compound, drug | Phosphatase Inhibitor Cocktail | Thermo Fisher | 78420 | |
| Chemical compound, drug | Poly-D-Lysine Solution | Sigma-Aldrich | A-003-M | |
| Chemical compound, drug | Protein A-Sepharose 4B | Thermo Fisher | 101042 | |
| Chemical compound, drug | Protein G-Sepharose 4B | Thermo Fisher | 101142 | |
| Chemical compound, drug | Proteinase Inhibitor Cocktail | Sigma-Aldrich | P8340 | |
| Chemical compound, drug | Sodium-hydrogen exchanger inhibitor | Merck KGaA | EMD87580 | |
| Chemical compound, drug | Sulfo-NHS-SS-biotin | Pierce | 21331 | |
| Chemical compound, drug | Tetrodotoxin | Sigma-Aldrich | T8024 | |
| Chemical compound, drug | Triton X-100 | Sigma-Aldrich | CAS9002-93-1 | |
| Chemical compound, drug | Vectashield with DAPI | Vector Labs | H-1200 | |
| Recombinant DNA reagent | pcDNA3.1-Zeo | Invitrogen | V79020 | |
| Recombinant DNA reagent | psPAX2 | Addgene | 12260 | |
| Recombinant DNA reagent | pMD2.G | Addgene | 12259 | |
| Recombinant DNA reagent | pLKO.1 | Addgene | 10878 | |

*Continued*

| Reagent type (species) or resource | Designation | Source or reference | Identifiers | Additional information |
|---|---|---|---|---|
| Recombinant DNA reagent | pLVXCMV100 | (*Dean et al., 2017*) | N/A | |
| Transfected construct (*Mus musculus*) | pCrl, Reelin expression vector | (*D'Arcangelo et al., 1997*) | N/A | |
| Transfected construct (*Mus musculus*) | pcDNA3.1-Apoer2-Fc | (*Hiesberger et al., 1999*) | N/A | |
| Transfected construct (*Homo sapiens*) | pcDNA3.1-ApoE2 | (*Chen et al., 2010*) | N/A | progenitor pcDNA3.1-Zeo |
| Transfected construct (*Homo sapiens*) | pcDNA3.1-ApoE3 | (*Chen et al., 2010*) | N/A | progenitor pcDNA3.1-Zeo |
| Transfected construct (*Homo sapiens*) | pcDNA3.1-ApoE4 | (*Chen et al., 2010*) | N/A | progenitor pcDNA3.1-Zeo |
| Transfected construct (*shRNA construct*) | pLKO.1-shRNA scramble | this paper | N/A | progenitor pLKO.1 |
| Transfected construct (*shRNA construct*) | pLKO.1-shNHE1 | Open Biosystem | TRCN0000044651 | progenitor pLKO.1 |
| Transfected construct (*shRNA construct*) | pLKO.1-shNHE5 | this paper | N/A | progenitor pLKO.1 |
| Transfected construct (*shRNA construct*) | pLKO.1-shNHE6 a | Open Biosystem | TRC N000 0068828 | progenitor pLKO.1 |
| Transfected construct (*shRNA construct*) | pLKO.1-shNHE6 b | Open Biosystem | TRCN0000068830 | progenitor pLKO.1 |
| Transfected construct (*shRNA construct*) | pLKO.1-shNHE6 c | Open Biosystem | TRCN0000068832 | progenitor pLKO.1 |
| Transfected construct (*shRNA construct*) | pLKO.1-shNHE7 | Open Biosystem | TRCN0000068812 | progenitor pLKO.1 |
| Transfected construct (*shRNA construct*) | pLKO.1-shNHE8 | this paper | N/A | progenitor pLKO.1 |
| Transfected construct (*shRNA construct*) | pLKO.1-shNHE9 | Open Biosystem | TRCN0000068856 | progenitor pLKO.1 |
| Transfected construct (*Mus musculus*) | pLVX-mCherry-Apoer2 | this paper | N/A | progenitor pLVXCMV100 |
| Transfected construct (*Homo sapiens*) | pcDNA3.1-ApoE3-GFP | this paper | N/A | progenitor pcDNA3.1-Zeo |
| Sequence-based reagent (oligo) | Scramble shRNA forward | IDT Inegrated DNA Technologies | N/A | 5'-CCGGCCTA AGGTTAA GTCGCCCT CGCTC-3' |

*Continued on next page*

*Continued*

| Reagent type (species) or resource | Designation | Source or reference | Identifiers | Additional information |
|---|---|---|---|---|
| Sequence-based reagent (oligo) | Scramble shRNA reverse | IDT Inegrated DNA Technologies | N/A | 5'-GAGCGAG GGCGACTT AACCTTAGG TTTTTG-3' |
| Sequence-based reagent (oligo) | shRNA anti NHE1 (SLC9A1) forward | Open Biosystem | TRCN0000044651 | 5'-CCGCCATC GGATCTT CCCTTCCT TACTCG-3' |
| Sequence-based reagent (oligo) | shRNA anti NHE1 (SLC9A1) reverse | Open Biosystem | TRCN0000044651 | 5'-AGTAAGGAAGGGAA GATCCGATGTTTTTG-3' |
| Sequence-based reagent (oligo) | shRNA anti NHE5 (SLC9A5) forward | IDT Inegrated DNA Technologies | N/A | 5'-CCGGAAG GACCACAC TCATCTTAG TCTCG-3' |
| Sequence based reagent (oligo) | shRNA anti NHE5 (SLC9A5) reverse | IDT Inegrated DNA Technologies | N/A | 5'-AGACTAAG ATGAGTG TGGTCCTTT TTTTG-3' |
| Sequence-based reagent (oligo) | shRNA anti NHE6 (SLC9A6) -a forward | Open Biosystem | TRCN0000068828 | 5'-CCGGGCCGTTTATA TGGCATAGGAACTC-3' |
| Sequence-based reagent (oligo) | shRNA anti NHE6 (SLC9A6)- a reverse | Open Biosystem | TRCN0000068828 | 5'-GAGTTCCTATGCCAT ATAAACGGCTTTTTG-3' |
| Sequence-based reagent (oligo) | shRNA anti NHE6 (SLC9A6)-b forward | Open Biosystem | TRCN0000068830 | 5'-CCGGCCCTTGTCTCT CTTACTTAATCTCG-3' |
| Sequence-based reagent (oligo) | shRNA anti NHE6 (SLC9A6)-b reverse | Open Biosystem | TRCN0000068830 | 5'-AGATTAAGTAAGAGA GACAAGGGTTTTTG-3' |
| Sequence-based reagent (oligo) | shRNA anti NHE6 (SLC9A6)-c forward | Open Biosystem | TRCN0000068832 | 5'-CCGGCCTTGGGTCT ATCTTAGCATACTCG-3' |
| Sequence-based reagent (oligo) | shRNA anti NHE6 (SLC9A6)-c reverse | Open Biosystem | TRCN0000068832 | 5'-AGTATGCTAAGATA GACCCAAGGTTTTTG-3' |
| Sequence-based reagent (oligo) | shRNA anti NHE7 (SLC9A7) forward | Open Biosystem | TRCN0000068812 | 5'-CCGGCCATTGTACT ATCCTCGTCTACTCG-3' |
| Sequence-based reagent (oligo) | shRNA anti NHE7 (SLC9A7) reverse | Open Biosystem | TRCN0000068812 | 5'-AGTAGACGAGGATA GTACAATGGTTTTTG-3' |
| Sequence-based reagent (oligo) | shRNA anti NHE8 (SLC9A8) forward | IDT Inegrated DNA Technologies | N/A | 5'-CCGGAAGGCTTCATG TGGTTGGATGCTC-3' |
| Sequence-based reagent (oligo) | shRNA anti NHE8 (SLC9A8) reverse | IDT Inegrated DNA Technologies | N/A | 5'-GAGCATCCAACCACAT GAAGCCTTTTTTTG-3' |
| Sequence-based reagent (oligo) | shRNA anti NHE9 (SLC9A9) forward | Open Biosystem | TRCN0000068856 | 5'-CCGGCTGGGCAGAAA GCAGAAGATTCTC-3' |
| Sequence-based reagent (oligo) | shRNA anti NHE9 (SLC9A9) reverse | Open Biosystem | TRCN0000068856 | 5'-GAGAATCTTCTGCTTT CTGCCCAGTTTTTG-3' |
| Sequence-based reagent (oligo) | Apoer2 NT cloning site (sdm) forward | Inegrated DNA Technologies | N/A | 5'-TACAAATCTAGAGATCCG CTGCCGGGCGGCCAAG-3' |
| Sequence-based reagent (oligo) | Apoer2 NT cloning site (sdm) reverese | Inegrated DNA Technologies | N/A | 5'-ACTCATGTCGACCGCTG CGGAGAGATG CTGAAGCTG-3' |
| Sequence-based reagent (oligo) | mCherry (for mCherry-Apoer2) forward | Inegrated DNA Technologies | N/A | 5'-AAATTCGTCGACATGGTG AGCAAGGGCGA GGAGGATAAC-3' |

*Continued on next page*

*Continued*

| Reagent type (species) or resource | Designation | Source or reference | Identifiers | Additional information |
|---|---|---|---|---|
| Sequence-based reagent (oligo) | mCherry (for mCherry-Apoer2) reverse | Inegrated DNA Technologies | N/A | 5'-GGGAACGTCTAGAG GACTTGTACAGCTC GTCCATG-3' |
| Sequence-based reagent (oligo) | Apoer2 forward | Inegrated DNA Technologies | N/A | 5'-TGGAGCGCTAGCGC CACCATGGGCCGCCC AGAACTGG-3' |
| Sequence-based reagent (oligo) | Apoer2 reverse | Inegrated DNA Technologies | N/A | 5'-AACCCGGAATTCTCA GGGCAGTCCAT CATCTTCAAGAC-3' |
| Sequence-based reagent (oligo) | NheI-site removal (sdm) forward | Inegrated DNA Technologies | N/A | 5'-GTTTACCGTCGA CCTCTAGCTAG-3' |
| Sequence-based reagent (oligo) | NheI-site removal (sdm) reverse | Inegrated DNA Technologies | N/A | 5'-AATGTCAAGGCCT CTCACTCTCTG-3' |
| Sequence-based reagent (oligo) | CMVfull forward | Inegrated DNA Technologies | N/A | 5'-CAGTTTATCGATG GCCAGATATACGCG TTGACATTG-3' |
| Sequence-based reagent (oligo) | CMVfull reverse | Inegrated DNA Technologies | N/A | 5'-TTTCCGCTAGCGGATCC CAGCTTGGGTCT CCCTATAGTGAGT-3' |
| Sequence-based reagent (oligo) | ApoE3 (ApoE3-GFP) forward( | Inegrated DNA Technologies | N/A | 5'-ATCAGGGAATTCAAC CATGAAGGTTCTG TGGGCTGCG-3' |
| Sequence-based reagent (oligo) | GFP (ApoE3-GFP) reverse | Inegrated DNA Technologies | N/A | 5'-ATTGGTGGATCCGCGT GATTGTCGCTG GGCACAG-3' |
| Software, algorithm | Adobe Creative Cloud | Adobe | RRID: SCR_010279 | |
| Software, algorithm | GraphPad Prism 7.0 | GraphPad Software | RRID: SCR_002798 | |
| Software, algorithm | Fiji/ImageJ | NIH | RRID: SCR_002285 | |
| Software, algorithm | LabView7.0 | National Instruments | RRID: SCR_014325 | |
| Software, algorithm | Odyssey Imaging System | LI-COR | RRID: SCR_014579 | |
| Software, algorithm | Clustal Omega | EMBL-EBI | RRID: SCR_001591 | |
| | Leica TCS SPE | Leica | RRID: SCR_002140 | |

## Contact for reagent and resource sharing

Request for reagents should be directed to Joachim Herz (Joachim.Herz@utsouthwestern.edu).

## Animals

ApoE3-KI or ApoE4-KI mice backcrossed to C57BL/6 were generously provided by Nobuyo Maeda and Patrick Sullivan (*Knouff et al., 1999*; *Sullivan et al., 1997*). Animals were group-housed on a standard 12 hr light/dark cycle and fed ad libitum standard mouse chow (Diet 7001; Harlan Teklad, Madison, WI, USA). Ethics Statement: All experimental procedures were performed according to the approved guidelines for Institutional Animal Care and Use Committee (IACUC) at the University of Texas Southwestern Medical Center at Dallas (Approval Number: A3472-01; 2015-101088).

## Human cortical brain extracts

Deidentified human cortical brain extracts (IRB exempt) from a non-AD, normal subject (Control) and a clinically and histopathologically confirmed Alzheimer's disease (AD) case were prepared as described previously (*Durakoglugil et al., 2009*). Briefly, control brain extract contained monomeric Aβ, but no detectable oligomers and only a trace amount of higher order aggregates; by contrast, AD brain extract contained monomeric Aβ in addition to Aβ dimers and higher order aggregates. One gram of brain tissue was homogenized in 4 ml of Tris buffered saline and centrifuged at 175,000 x g as described by *Shankar et al. (2008)*. The supernatant was designated 'extract'. The genotype of the brain tissue was ApoE3/3 for the control and ApoE3/4 for the AD tissue.

## Primary neurons

Primary rat (Sprague-Dawley) cortical neurons (E18) were prepared as described previously (*Chen et al., 2005*) and cultured in six-well plates (1 million neurons/9 $cm^2$) or on Poly-D-Lysine coated coverslips (30,000 neurons/1.1 $cm^2$) in Neurobasal/B27 medium at 37°C and 5% $CO_2$. At 9–14 days in vitro (DIV) primary neurons were used for experiments.

## Plasmid constructs

All primers used for cloning are listed in the Key Resources Table.

*pcDNA3.1-Apoer2-Fc:* The mouse ApoER2-Fc construct (secreted Apoer2 ectodomain), tagged with the V5 epitope and Fc, was described previously (*Hiesberger et al., 1999*). To produce recombinant Apoer2-Fc, HEK 293 cells were transfected and the medium was harvested as described (*Chen et al., 2010*).

*pLKO.1-shRNA constructs:* pLKO.1-constructs containing shRNA targeting different NHE subtypes were purchased from Sigma or created by inserting aligned oligos (for shRNA sequence refer to Key Resources Table) into the pLKO.1-TRC, as described elsewhere (*Moffat et al., 2006*). *pLVX-mCherry-Apoer2:* SalI and XbaI cloning sites were inserted into a plasmid containing the Apoer2 full-length cDNA immediately downstream of the Apoer2 signal peptide (N-terminus, NT) by site-directed mutagenesis. In a second step, PCR-amplified mCherry was inserted into the newly created SalI and XbaI sites. NT-mCherry-Apoer2 was then amplified by PCR and cloned into the NheI and EcoRI sites of lentiviral vector that was generated by modifying pLVXCMV100 by removing an NheI site through site-directed mutagenesis and replacing the truncated CMV100 with a full length CMV promoter (inserted into the ClaI and NheI sites). *pcDNA3.1-ApoE3-GFP:* ApoE3 was PCR-amplified and inserted into pEGFP-N1 using EcoR1 and BamHI cloning sites.

## Generation of recombinant proteins

### Reelin

HEK 293 cells stably expressing Reelin (pCrl) were cultured in DMEM medium (*D'Arcangelo et al., 1995*; *Förster et al., 1998*). Medium containing Reelin was harvested and purified as described previously (*Weeber et al., 2002*). Authenticated by their ability to secrete Reelin. Mycoplasma status ascertained annually, last negative result after submission of this manuscript.

The ApoE used in this study was contained in lipoprotein particles naturally secreted from transfected cells in a minimally lipidated state and produced as follows: HEK 293 cells were transfected (FuGENE) with pcDNA3.1 vector containing full length human ApoE (ApoE2, ApoE3, ApoE4, ApoE4P, mutant ApoE3, or mutant ApoE4) cDNA or empty pcDNA3.1 (control). 24 hr post-transfection DMEM medium was replaced with Neurobasal medium. After 3 days, the medium containing the different ApoE isoforms was collected and analyzed on SDS-PAGE to calculate ApoE concentrations using commercial human ApoE3 as a standard. The proteins were immunoblotted for ApoE and detected with IRDye 800CW secondary antibody (Li-Cor). ApoE levels were quantified Odyssey infrared imaging.

## Western blots

After treatment, neurons were washed three times with cold PBS, and lysed in RIPA buffer (50 mM Tris-HCl, pH 8.0; 150 mM NaCl; 1% Nonidet P-40; phosphatase and protease inhibitors) for 20 min on ice. Cellular debris was removed by centrifugation for 10 min at 14,000 rpm and 4°C in an Eppendorff centrifuge. Protein concentrations were measured using the Bradford Protein Assay (Bio-Rad).

After adding 4x SDS loading buffer (0.1 M Tris-HCl, pH 6.8, 2% SDS, 5% β-mercaptoethanol, 10% glycerol, and 0.05% bromphenol blue) the samples were boiled at 95°C for 10 min. For immunoblotting of NHE6, samples were incubated for 30 min at room temperature instead of boiling. 4x SDS loading buffer was added to the media to detect secreted proteins. After boiling (95°C for 10 min) samples were loaded on SDS-PAGE. Proteins were transferred to a nitrocellulose membrane for western blotting with the indicated antibodies.

## Confocal microscopy

For imaging DIV9 neurons on 12 mm coverslips were infected with lentiviral DNA encoding NT-mCherry-Apoer2. Lentivirus containing medium was removed 14 hr after infection. On DIV12 ApoE3-GFP containing supernatant of 293 cells transfected with pcDNA3.1-ApoE3-GFP (FuGENE) was added. On DIV13 neurons were washed 2x with PBS and fixed with 4% PFA. Coverslips were mounted using Vectashield Antifade Mounting Medium with DAPI. Z-stack images were obtained using a Confocal Zeiss LSM880 Airyscan microscope and a 63x objective and a step size of 1 μm. 3D projections and orthogonal views were generated using NIH Fiji/ImageJ software.

## Surface biotinylation/Apoer2 recycling

Primary neurons were pre-treated for 30 min with ApoE-conditioned medium (5 μg/ml unless stated differently), then incubated with Reelin (2 μg/ml) for an additional 30 min (see timeline in *Figure 2A*). After treatment cells were washed with cold PBS and incubated in PBS containing sulfo-NHS-SS-biotin (1.0 mg/ml) for 30 min at 4°C. Excess reagent was quenched by rinsing the neurons with cold PBS containing 100 mM glycine. Neurons were lysed in 160 μl/9 cm$^2$ lysis buffer (PBS with 0.1% SDS, 1% Triton X-100, and protease inhibitors) at 4°C for 20 min. Cell debris was removed by centrifugation for 10 min at 14,000 rpm at 4°C in an Eppendorff centrifuge. The protein concentration was measured using the Bradford Protein Assay (Bio-Rad) and 100 μg of total protein was incubated with 50 μl of NeutrAvidin agarose at 4°C for 1 hr. Agarose pellets were washed three times using washing buffer (500 mM NaCl; 15 mM Tris-HCl, pH 8.0; 0.5% Triton X-100), biotinylated surface proteins were eluted from agarose beads by boiling in 2x SDS sample loading buffer and loaded on SDS-PAGE for western blot analysis. For drug treatments, cells were pre-incubated with EMD87580 and/or Bafilomycin for 1 hr prior to ApoE and Reelin addition. 3 μM EMD87580 (3 mM stock in PBS) was used unless stated differently.

## Co-immunoprecipitation

Apoer2-Fc/ApoE interaction: Protein G Sepharose (50 μl) was added to 1 ml culture supernatant containing ApoER2-Fc from HEK 293 cells at 4°C overnight. Beads were then sedimented by brief centrifugation, and then 500 μl culture supernatant containing ApoE and CaCl$_2$ (final concentration 1 mM) was added to the beads. The mixture was incubated for an additional 4 hr at 4°C. Beads were washed three times using washing buffer (500 mM NaCl, 15 mM Tris.HCl, 0.5% Triton X-100 (pH8.0)), and bound proteins were separated on 4 – 15% SDS-PAGE and immunoblotted for ApoE.

Apoer2-ApoE interaction: ApoE-treated (5 μg/ml for 3 hr at 37°C) neurons were washed with ice cold PBS and lysed in RIPA buffer. For immunoprecipitation 600 μg of lysate were co-incubated with anti-Apoer2 rabbit serum or control rabbit serum and protein A-Sepharose beads at 4°C overnight. Precipitated beads were washed 3x in RIPA buffer, resuspended in 2x SDS sample buffer and boiled at 95°C for 10 min. Eluted proteins were probed for Apoer2 and ApoE.

## Dab1 phosphorylation

Primary neurons were treated with Reelin (2 μg/ml for 30 min) or ApoE3 or ApoE4 at 37°C for 3 hr. After treatment, neurons were lysed and prepared for western blotting as described above and probed with antibodies raised against total Dab1 and phospho-Tyrosine (4G10) to identify phospho-Dab1 (*Masereel et al., 2003*).

## Preparation of lentiviral particles

HEK 293 T cells were co-transfected with psPAX2, pMD2.G, and the individual transfer constructs (pLKO.1-shRNA or pLVX-mCherry-Apoer2). Media was replaced after 12 – 15 hr. Viral particle containing media was collected and cell debris spun down. The virus was concentrated by ultra-

centrifugation and re-suspension in DMEM (1/10$^{th}$ volume). To transfect neurons 100 µl of concentrated virus was added per ml of medium in the culture dish. Transduction media were replaced 12 – 15 hr after infection and neurons were incubated for 3 days before experiments were conducted.

### In vivo EMD87580 treatment

For in vivo treatment, mice were intraperitoneally injected with EMD87580 (1 mg/ml in PBS) at a dose of 5 mg/kg. Additionally, mice received intranasal application of 10 µl of 1 mg/ml EMD87580. Animals were treated twice a day for 2 consecutive days (in the morning and evening, 12 hr interval). On the third day, mice were treated with EMD87580 in the morning and sacrificed 2 hr later for extracellular field recordings of hippocampal slices.

### Extracellular field recordings

Hippocampal slices were prepared from 2 to 3 months old ApoE3-KI or ApoE4-KI mice. Brains were quickly removed and placed in cold high sucrose cutting solution (110 mM sucrose, 60 mM NaCl, 3 mM KCl, 1.25 mM NaH$_2$PO$_4$, 28 mM NaHCO$_3$, 0.5 mM CaCl$_2$, 5 mM glucose, 0.6 mM Ascorbic acid, 7 mM MgSO$_4$). 400 µm transverse sections were cut using a vibratome. Slices were then transferred into an incubation chamber containing 50% aCSF (124 mM NaCl, 3 mM KCl, 1.25 mM NaH$_2$PO$_4$, 26 mM NaHCO$_3$, 10 mM D-glucose, 2 mM CaCl$_2$, 1 mM MgSO$_4$) and 50% sucrose cutting solution. For electrophysiological recordings a final concentration of 3 µM EMD87580 was used. For combined treatment with Reelin, AD brain extract and EMD87580 (*Figure 8*), slices were pretreated with EMD87580 for 3 hr and that concentration was maintained during recording in the presence or absence of Reelin and/or AD extract. Slices were transferred to the interface recording chamber where they were kept in aCSF at 31°C and a flow rate of 2 – 3 ml/min. In the recording chamber, different combinations of treatment with AD brain extract, Reelin, and EMD87580 were used. Slices were perfused with the different final components for an additional ~30 min until they stabilized in the recording chamber before application of theta burst stimulation and one hour thereafter throughout the recording period. For stimulation concentric bipolar electrodes were used (FHC, Catalog no CBBRC75, 1201 Main St Bowdoin, ME 04287, USA) and placed into the stratum radiatum. Stimulus intensity was set at 40 – 60% maximum response and delivered through an Isolated Pulse Stimulator (A-M Systems, Model 2100. A custom written program in Labview 7.0 was used for recording and analysis of LTP experiments (courtesy of Dr Jay Gibson). A theta burst (TBS; train of 4 pulses at 100 Hz repeated 10 times with 200 ms intervals and again repeated 5 times at 10 s intervals) was used as conditioning stimulus. For input/output analysis data were binned and fitted linearly. Slopes were calculated using regression analysis.

### Quantification and statistical analysis

Data were expressed as the mean ± SEM and evaluated using two-tailed Student's t test for two groups with one variable tested and equal variances, or one-way analysis of variance (ANOVA) with Dunnett's post-hoc for multiple groups with only variable tested. The differences were considered to be significant at $p < 0.05$ (*$p < 0.05$, **$p < 0.01$, ***$p < 0.001$).

### Acknowledgements

This work was supported by NIH grant R37 HL63762, R01 NS093382, R01 NS108115, and RF1 AG053391 (to JH). While this work was conceived or ongoing, JH is or was further supported by the Alexander-von-Humboldt Foundation, the American Health Assistance Foundation, the Consortium for Frontotemporal Dementia Research, the Bright Focus Foundation, the Lupe Murchison Foundation, and The Ted Nash Long Life Foundation. The early stages of this work were partially supported by an unrestricted research grant from Merck KGaA. We are indebted to Rebekah Hewitt, Huichuan Reyna, Issac Rocha, Tamara Terrones, Emily Boyle, Alisa Gilloon, Travis Wolff, and Eric Hall for their excellent technical assistance, and Wolfgang Scholz and Dirk Beher for sharing reagents. Nancy Heard and Barbara Dacus for help with art work. Image acquisition was supported by NIH grant 1 S10 OD021685-01A1 to Kate Luby-Phelps. We also thank Charles White and the ADC at UT Southwestern for providing the pathology specimens first cited in (Durakoglugil et al., 2009), and Dr. Yuan Yang for performing several of the early experiments that were the basis for this paper. The sole

reason why she is not included as a coauthor is that she has returned to China without leaving a contact address. Therefore, we were unable to provide the necessary assurance that she has read and approved the paper. She should be able to claim this publication as a coauthor and we would be delighted to add her as a third contributing author.

## Additional information

### Funding

| Funder | Grant reference number | Author |
|---|---|---|
| National Institutes of Health | R37 HL63762 | Joachim Herz |
| BrightFocus Foundation | A2016396S | Joachim Herz |
| Bluefield Project | | Joachim Herz |
| National Institutes of Health | R01 NS108115 | Joachim Herz |
| National Institutes of Health | R01 NS093382 | Joachim Herz |
| National Institutes of Health | RF1 AG053391 | Joachim Herz |

The funders had no role in study design, data collection and interpretation, or the decision to submit the work for publication.

### Author contributions

Xunde Xian, Conceptualization, Data curation, Formal analysis, Validation, Investigation, Visualization, Methodology, Writing—original draft, Writing—review and editing; Theresa Pohlkamp, Data curation, Formal analysis, Validation, Investigation, Visualization, Methodology, Writing—review and editing; Murat S Durakoglugil, Data curation, Formal analysis, Validation, Investigation, Methodology; Connie H Wong, Data curation, Formal analysis, Validation, Investigation, Methodology, Writing—review and editing; Jürgen K Beck, Conceptualization, Funding acquisition; Courtney Lane-Donovan, Resources, Data curation, Investigation, Methodology; Florian Plattner, Formal analysis, Supervision, Validation, Project administration, Writing—review and editing; Joachim Herz, Conceptualization, Resources, Formal analysis, Supervision, Funding acquisition, Validation, Investigation, Methodology, Writing—original draft, Project administration, Writing—review and editing

### Author ORCIDs

Xunde Xian (iD) https://orcid.org/0000-0003-3059-1254
Murat S Durakoglugil (iD) http://orcid.org/0000-0003-4483-8166
Connie H Wong (iD) http://orcid.org/0000-0002-6452-7966
Courtney Lane-Donovan (iD) http://orcid.org/0000-0001-9504-8346
Florian Plattner (iD) http://orcid.org/0000-0002-3150-1866
Joachim Herz (iD) http://orcid.org/0000-0002-8506-3400

### Ethics

Animal experimentation: All experimental procedures were performed according to the approved guidelines for Institutional Animal Care and Use Committee (IACUC) at the University of Texas Southwestern Medical Center at Dallas (Approval Number: A3472-01).

### Decision letter and Author response

Decision letter https://doi.org/10.7554/eLife.40048.023
Author response https://doi.org/10.7554/eLife.40048.024

## Additional files

### Supplementary files

• Transparent reporting form

DOI: https://doi.org/10.7554/eLife.40048.021

**Data availability**

All data generated or analysed during this study are included in the manuscript and supporting files.

---

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
