## [Decision Letter]

Thank you for submitting your article "Reversal of ApoE4 Induced Recycling Block as a Novel Prevention Approach for Alzheimer's Disease" for consideration by *eLife*. Your article has been reviewed by two peer reviewers, and the evaluation has been overseen by a Reviewing Editor and Huda Zoghbi as the Senior Editor. The following individuals involved in review of your submission have agreed to reveal their identity: Eric Morrow (Reviewer #1).

The reviewers have discussed the reviews with one another and the Reviewing Editor has drafted this decision to help you prepare a revised submission.

Summary:

This manuscript presents a new model of the action of the Alzheimer's disease risk variant ApoE4. Xian et al. describe a role for the proton leak channel, NHE6, in the ApoE4-induced block in endocytic recycling of Apoer2 (LRP8) in cultured neurons. The authors demonstrate that Apoer2 is endocytosed concomitantly with ApoE, but that ApoE4 causes Apoer2 to be retained in endosomes, alters the pH of endosomes, and blocks the recycling of AMPA and NMDA. To reverse this inhibition, the authors blocked and reduced levels of NHE6, the proton leak channel in the early endosome, via RNAi and pharmacological inhibition. NHE6 inhibition decreased the endosomal pH and reestablished ApoE4 stability and localization to the plasma membrane as well as increased Apoer2, AMPA, and NMDA glutamate receptor recycling.

While this manuscript provides a clearer picture of the endocytic defects associated with ApoE4 using cultured neurons, the authors need to make the following revisions to the manuscript prior to acceptance.

Essential revisions:

1) The authors should clarify whether the images in Figure 1C are a single slice or a composite of a Z-stack to demonstrate that punctae in the ApoE channel are indeed in the same focal plane as those in Apoer2 channel. The authors should further clarify in the Materials and methods section how these images were captured and how the co-localization was established.

2) In Figures 4B and C, the text states that ApoE3 impaired Apoer2 recycling; there are data only for ApoE4. The authors should either include the ApoE3 data or remove that statement.

3) From Figures 7A and B, EMD87580 treated ApoE4-KI tissue showed less LTP than control ApoE4-KI tissue, which should be noted in the text. Based on data with Reelin treatments, the authors interpret that, in ApoE4-KI tissue, Reelin only has an effect in the presence of EMD87580. However, this comparison does not factor in the effects of EMD87580 alone. The authors should provide further analysis to test their model, or discuss their findings in the context of an effect of EMD87580 alone on LTP. The effects of Reelin and NHE6 inhibition on synaptic modulation is a major finding (summarized in the Abstract) and thus important to be well documented.

4) In Figure 7D, the authors show paired data of tissue treated with Reelin and tissue treated with EMD87580 and Reelin, but without justification of why it is appropriate to show data as paired samples. It should be clarified in the figure legend that these data are derived from Figures 7A and B.

5) In Figure 8D, the authors demonstrate that 30 minutes of EMD87580 was insufficient to normalize LTP but 4 hours was sufficient. They conclude that the four hours are necessary to alleviate the ApoE4-mediated Reelin resistance. However, there are not statistically significant differences between the responses at 30 minutes and four hours. The conclusion should be left out of the Results, although perhaps considered in the Discussion.

---

## [Author Response]

Essential revisions:1) The authors should clarify whether the images in Figure 1C are a single slice or a composite of a Z-stack to demonstrate that punctae in the ApoE channel are indeed in the same focal plane as those in Apoer2 channel. The authors should further clarify in the Materials and methods section how these images were captured and how the co-localization was established.

We thank the reviewer for making us aware of this lack of information. Figure 1C shows a single plane of a Z-stack, we added this information to the text. For completeness we added an orthogonal view (3D presentation) to Figure 1 and the information about the Z-stack spacing to the legend. We also provide 3D videos as supplemental material.

2) In Figures 4B and C, the text states that ApoE3 impaired Apoer2 recycling; there are data only for ApoE4. The authors should either include the ApoE3 data or remove that statement.

The statement referred to the data shown in Figures 4D and E. We apologize for this oversight; we have corrected it and now refer to Figures 4D and E.

3) From Figures 7A and B, EMD87580 treated ApoE4-KI tissue showed less LTP than control ApoE4-KI tissue, which should be noted in the text. Based on data with Reelin treatments, the authors interpret that, in ApoE4-KI tissue, Reelin only has an effect in the presence of EMD87580. However, this comparison does not factor in the effects of EMD87580 alone. The authors should provide further analysis to test their model, or discuss their findings in the context of an effect of EMD87580 alone on LTP. The effects of Reelin and NHE6 inhibition on synaptic modulation is a major finding (summarized in the Abstract) and thus important to be well documented.

We thank the reviewer for helping us to clarify this important point. For completeness we have now added data for ApoE3-KI animals to Figure 7. We have accordingly added the description to the Results to address the effect of EMD87580 alone on ApoE3-KI and ApoE4-KI. We have also added a whole paragraph to the Discussion where we have put the results of these field recordings into the appropriate context with previously published data by others, which further supports our results.

4) In Figure 7D, the authors show paired data of tissue treated with Reelin and tissue treated with EMD87580 and Reelin, but without justification of why it is appropriate to show data as paired samples. It should be clarified in the figure legend that these data are derived from Figures 7A and B.

Again, we thank the reviewer for pointing this out. We indeed did not describe the exact protocol of how the pairing in the experiment was designed. We paired the different treatments of slices derived from the same brain as an additional control to exclude a systematic error of the setup. Thus, the data shown in the old Figure 7D panel were a reinterpretation of the data shown in the old Figures 7A and B. To avoid confusion, and since these data are not important for the interpretation, we removed Figure 7D from the manuscript. Also please note, as mentioned in our answer to point 3, the new Figure 7 now includes data from ApoE3-KI slices previously not included (new Figure 7A) next to the ApoE4-KI slice recordings (new Figure 7B). The input-output graphs are shown in the new Figures 7C and D, respectively.

5) In Figure 8D, the authors demonstrate that 30 minutes of EMD87580 was insufficient to normalize LTP but 4 hours was sufficient. They conclude that the four hours are necessary to alleviate the ApoE4-mediated Reelin resistance. However, there are not statistically significant differences between the responses at 30 minutes and four hours. The conclusion should be left out of the Results, although perhaps considered in the Discussion.

We agree with the reviewer’s suggestions, and we removed the 30 minute time point from Figure 8. This is now merely stated as an ancillary observation not shown.